green chemistry/plant science/analytical chemistry

*Ixora javanica*, flavonoid, response surface methodology, deep eutectic solvent, ultrasound-assisted extraction

**Author for correspondence:**
Abdul Mun'im
e-mail: munim@farmasi.ui.ac.id

# A green extraction design for enhancing flavonoid compounds from *Ixora javanica* flowers using a deep eutectic solvent

Nina Dewi Oktaviyanti[1,3], Kartini Kartini[3], Mochammad Arbi Hadiyat[4], Ellen Rachmawati[3], Andre Chandra Wijaya[3], Hayun Hayun[2] and Abdul Mun'im[1,2]

[1]Department of Pharmacognosy-Phytochemistry, Faculty of Pharmacy, and [2]Graduate Program of Herbal Medicine, Faculty of Pharmacy, Universitas Indonesia, Kampus UI, Depok 16424, West Java, Indonesia
[3]Department of Pharmaceutical Biology, Faculty of Pharmacy, and [4]Department of Industrial Engineering, Faculty of Engineering, Universitas of Surabaya, Surabaya 60293, East Java, Indonesia

NDO, 0000-0003-0134-0544; KK, 0000-0002-7685-9933; MAH, 0000-0003-0215-0259; HH, 0000-0002-1495-6228; AM, 0000-0002-6681-9196

In this study, an environmentally friendly extraction method for flavonoid compound from *Ixora javanica*, as a new raw material candidate for herbal medicine and cosmetics, was developed. The objectives of the present work were to provide recommendations for the optimal extraction conditions and to investigate the effects of any extraction parameters on flavonoid yields from the *I. javanica* flower. The extraction process was performed using deep eutectic solvent (DES) (choline chloride and propylene glycol at molar ratio of 1:1) and the ultrasound-assisted extraction method. Both single-factor and response surface analyses using three-level and three-factor Box Behnken designs were conducted to obtain the optimum flavonoid concentrations. The results showed that the optimum extraction conditions for total flavonoids featured an extraction time of 40 min, 25% water content in DES and a solid-to-liquid ratio of 1:25 g ml$^{-1}$. An extract obtained under optimum extraction conditions showed higher total flavonoid yields than an ethanolic extract which was used for comparison. Scanning electron microscope images demonstrated that both of the solvents also showed different effects on the outer surface of the *I. javanica* flower during the extraction process. In summary, our

work succeeded in determining the optimum conditions for total flavonoids in the *I. javanica* flower using a green extraction method.

# 1. Introduction

Flavonoids are the most abundant secondary metabolites and are known to be responsible for various biological activities in humans. At present, flavonoids have become a concern because of their benefits against many degenerative diseases, such as cardiovascular disease, cancer and diabetes. Many researchers have correlated the various activities of flavonoids with their very strong antioxidant activity [1–3]. Therefore, plants with a high level of flavonoids are prospective for development as herbal preparations to treat degenerative and age-related disease.

Various attempts have been made to obtain optimum flavonoid compounds from plants. Non-conventional extraction methods, such as ultrasound-assisted extraction (UAE), are often applied to improve extraction efficiency and increase bioactivity. In addition, these methods have many advantages, including environmental friendliness, the need for minimal solvent and time efficiency [4,5]. The development of flavonoids and other plant metabolite extraction processes is not only limited to the use of organic solvents. Many previous studies have applied deep eutectic solvents (DESs) as alternative environmentally friendly solvents for the extraction of flavonoids [6–8]. Generally, DESs have been widely used to replace organic solvents. The use of DESs as a green solvent in the extraction process is currently highly prospective because DESs are very simple, cheap, less toxic, have easy preparation, and can be adjusted according to the purpose of extraction [9,10]. Moreover, some hydrogen bond acceptors (HBA) and hydrogen bond donors (HBD) are also excipients in the formulation of medicinal and cosmetic products. Thus, the extract containing DES has potential to be formulated directly without going through a solvent separation process first. It can be used as a strategy to reduce the process of energy-consumption and excipients used.

*Ixora javanica*, an attractive red flowering plant that is a member of the Rubiaceae family, is known to contain high levels of flavonoid compounds, especially in its flowers. Phytochemical studies of *I. javanica* reported the presence of flavonoids such as quercetin, formononetin and anthocyanin [11,12]. Besides flavonoid content, the flowers were also reported to contain many phenolic and terpenoid compounds, which are responsible for many activities, such as antioxidant, anti-tumour, anti-inflammatory, hepatoprotective and tyrosinase-inhibiting activities [13–16].

To date, studies related to the application of DES and UAE for *I. javanica* flower extraction are limited. In our previous study, we were able to optimize the extraction variables, including extraction time, solid-to-liquid ratio and temperature, to obtain the maximum flavonoid compounds using the response surface method [17]. To the best of our knowledge, there are no studies reporting the optimum level of water in DES needed to increase the effectivity of DES in flavonoid compound extraction from *I. javanica* flowers. Therefore, in this present work, we optimized the extraction conditions by using water content in DES as one of the extraction variables, followed by extraction time and the solid-to-liquid ratio. Unlike the previous study, here, we performed the extraction at room temperature.

The main purpose of this study is not only to provide recommendations for the optimum extraction conditions using the response surface method but also to investigate the effect of each extraction variable on flavonoid compound yields by using single-factor experiments. A single-factor analysis was first employed in a preliminary study before optimization using the response surface method. To offer a better explanation of extraction efficiency, we also performed scanning electron microscopy (SEM) imaging on dried flower powder.

# 2. Material and methods

## 2.1. Chemicals and materials

Choline chloride was purchased from Xi'an Rongsheng Biotechnology Co, Ltd, China, while other solvents such as propylene glycol and ethanol were acquired from Merck, Germany. The quercetin standards used in this study were purchased from Sigma Aldrich, USA.

## 2.2. Plant materials

The fresh red coloured the *I. javanica* flower used in this study was authenticated by the Center for Traditional Medicine Information and Development, Faculty of Pharmacy, University of Surabaya.

The flower was collected at full bloom from the Tenggilis Mejoyo District, Surabaya, East Java, Indonesia. After sorting, the flowers were washed, shade dried and powdered. The dry powder was then sieved through a size 30 mesh sifter, stored in a sealed container and kept under dry environmental conditions.

## 2.3. Preparation of deep eutectic solvent

In brief, DESs are prepared from at least two components (e.g. HBA and HBD) at an appropriate molar ratio. The DES components used in this study were the combination of choline chloride and propylene glycol (ChCl-Pg). DESs were prepared by heating and stirring both components at certain molar ratios (1 : 1; 1 : 2; 2 : 1) in a flask at 50°C for 30 min constantly until a homogeneous clear mixture formed. Deionized water was added to the DES to prepare the DES solution containing various concentrations of water. The water content used in this study was 5%–35% w/w of water in the DES. All DES mixtures obtained in this study were stable in liquid form under room temperature storage. Viscosity of DES mixtures were measured before being used for the extraction process (Brookfields™ cone plate viscometer, Ametek Brookfield, Middleboro, USA).

## 2.4. Extraction procedures

The extraction of *I. javanica* flowers was carried out using the UAE method at room temperature. When investigating the effect of one variable on the flavonoid yields, the other variables were kept constant. To determine the effect of water content in DES on the flavonoid yields, 0.5 g dried powder was extracted with 10 ml DES (ChCl-Pg at a molar ratio of 1 : 1), which contained different concentrations of water in DES (5%, 10%, 15%, 20%, 25%, 30% and 35% w/w) for 5 min. Extraction time: 0.5 g dried powder was mixed with 10 ml DES (ChCl-Pg at a molar ratio of 1 : 1 and water content of 20% w/w), and the extraction was carried out under different extraction times (10, 15, 20, 25, 30, 35 and 40 min). The solid-to-liquid ratio: 0.5 g dried powder was mixed with different volumes of DES (ChCl-Pg at a molar ratio of 1 : 1 and water content of 20% w/w) until reaching a certain solid-to-liquid ratio (1 : 23, 1 : 24, 1 : 25, 1 : 26, 1 : 27, 1 : 28 and 1 : 29 g ml$^{-1}$) and was then sonicated for 5 min. The ultrasonic bath was set at room temperature with a fixed frequency radiation of 40 kHz. The extracts obtained were then centrifuged at 1500 rpm for 15 min, and the filtrates were collected. The filtrate was then analysed for its flavonoid levels. Extraction using a conventional extraction solvent was carried out by sonicating 0.5 g dried material using ethanol under the optimum solid-to-liquid ratio and extraction time obtained from the response surface methodology (RSM) analysis. The flavonoid levels in the ethanolic extract were used as a comparison with those in the DES extracts. All procedures of extraction in this study were conducted in triplicate.

## 2.5. Determination of total flavonoids

The total levels of flavonoids in the extracts were determined using the method applied in the previous study with very slight modifications [17]. About 1.0 ml of each extract's filtrates was mixed with 1.5 ml of 0.33% AlCl$_3$ and 1.5 ml of 10% acetic acid solution. The mixtures were added with 96% ethanol until reaching a total volume of 10.0 ml. After 30 min incubation, the mixed solutions were analysed spectrophotometrically (UV-1900, Shimadzu Corp, Kyoto, Japan) at 425.8 nm. Quercetin was used as a standard compound so that the total flavonoid levels in the dried flower powder could be expressed as their quercetin equivalent (mg QE g$^{-1}$ dried flowers). Total flavonoid was calculated using a standard curve of prepared quercetin solution ranging from 3.5 to 11.5 mg ml$^{-1}$ with $y = 0.0678x + 0.0257$ ($R^2 = 0.999$). All of the analysis procedures were conducted in triplicate.

## 2.6. Extraction optimization using response surface methodology

To optimize the extraction process of the *I. javanica* flower to obtain the highest total flavonoid level, three independent variables, each of which consisted of three levels, were statistically analysed by a Box Behnken design (BBD) using the RSM. These three levels of variables were obtained from the analysis of each extraction parameter using a single factor. The BBD was used to formulate a second-order polynomial model which expressed total flavonoid yields:

$$Y = \beta_0 + \sum_{j=1}^{3} \beta_j X_j + \sum_{j=1}^{3} \beta_{jj} X_j^2 + \sum_{i=1}^{2} \sum_{j=i+1}^{3} \beta_{ij} X_i X_j,$$

**Table 1.** The code, range and level of each variable selected for the experimental design.

| variables | unit | code | range and level (xi) | | |
|---|---|---|---|---|---|
| | | | −1 | 0 | 1 |
| time | min | $X_1$ | 20 | 30 | 40 |
| water content in DES | % | $X_2$ | 15 | 25 | 35 |
| solid-to-liquid ratio | g ml$^{-1}$ | $X_3$ | 1/25 | 1/26 | 1/27 |

where $Y$ is the response variable (total flavonoid yields); $\beta_0$ is a constant and represents the intercept; $\beta_j$, $\beta_j$, $\beta_{ij}$ are the linear, squared and interaction coefficients, respectively [3]. All variables and selected levels selected are represented in table 1. A total of 15 runs were generated according to the software [18].

## 2.7. Scanning electron microscopy

Dried flower powder, both before and after the extraction process, was coated with 5 nm of Au before the imaging process using SEM (Inspect S50, Fei, Japan) at magnifications of 500× and 5000×. The SEM imaging was performed to visualize the surfaces of the solid materials and correlate then with the extraction efficiency.

## 2.8. Data analysis

In this study, the effects of extraction parameters on flavonoid yields were analysed via a single-factor assisted one-way analysis of variance (ANOVA) test ($p < 0.05$ significance level; SPSS software v. 18 for Windows, IBM, New York, United States) and the response surface method (Design Expert Software, v. 11, Stat-Ease Inc., Minneapolis, MN, USA). All total flavonoid data in the text and tables are presented as the mean ± standard deviation (s.d.); additionally, the s.d. is indicated as error bars in the figures.

# 3. Results and discussion

## 3.1. Effect of different extraction variables on flavonoid yields

### 3.1.1. Viscosity and water content in deep eutectic solvent

The selection of the extraction solvent is crucial to increase the acquisition of the target compounds. There are many kinds of HBA and HBD that can be used for DES preparation [19,20]. A combination of HBAs and HBDs in the DES preparation can affect the characteristics, polarity and dissolving ability of DESs [21–23]. The selection of HBA and HBD should be adjusted to the target compound according to the 'like dissolve like' principle. Our previous study reported that a combination of choline chloride as an HBA and propylene glycol as an HBD offers the most effective flavonoid extraction from *I. javanica* flowers out of 11 other types of DES [17]. In the extraction process, the viscosity of DES is a common problem [24]. A high viscosity extraction solvent makes mass transfer difficult and decreases the extraction yields [25,26]. Increasing the extraction temperature can decrease viscosity [27]. Theoretically, the desirable extraction temperature using DES ranges from room temperature to about 60°C. However, if the temperature keeps increasing, the yields of the target substance that are thermally sensitive may be decreased. Moreover, the use of a high temperature in extraction may increase the energy consumption, which is contradictive to the green extraction principle [28]. A study conducted by Dai *et al.* [29] highlighted that the extraction of flavonoid compounds can be carried out at ambient temperatures and provide the maximum extraction recovery for flavonoid compounds. Therefore, in this study, we decided to carry out the extraction process at room temperature, unlike our previous studies.

In the present study, we added water to the DES mixture to solve the viscosity problems of DES as has been conducted by several studies [10,30,31]. Our results demonstrated that water addition resulted in a lower viscosity of DES. Figure 1 shows that adding a greater amount of water into DES resulted in higher extraction yields, which reached the maximum flavonoid yield of 68.985 mg QE g$^{-1}$ dried flowers at a water concentration of 25%. Furthermore, the addition of water above 25% subsequently decreased

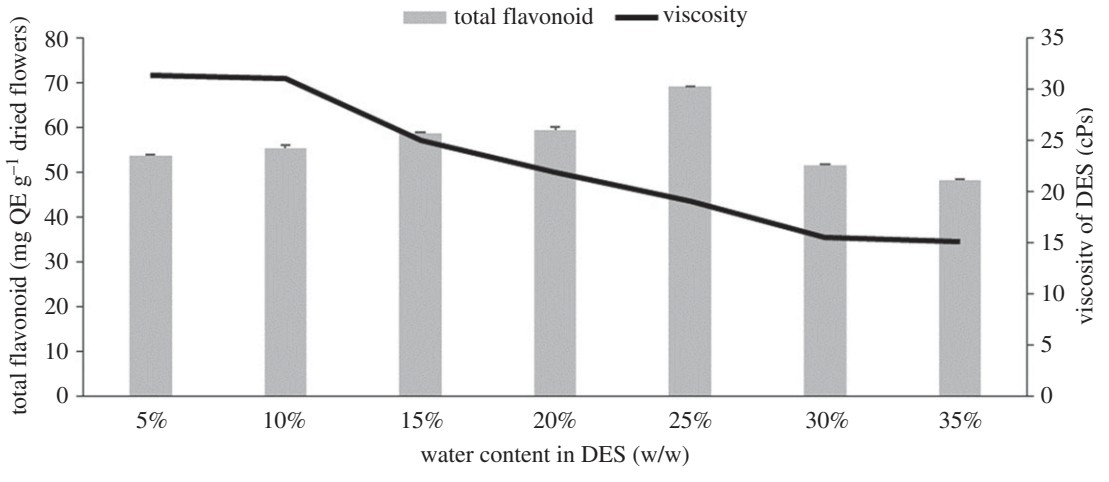

**Figure 1.** The effect of different water quantities in DES on flavonoid yields.

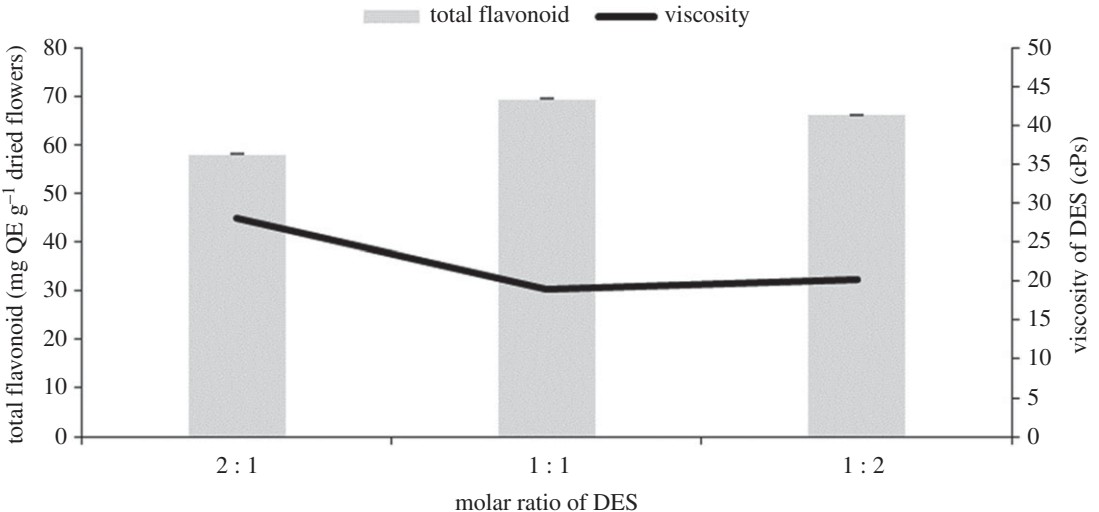

**Figure 2.** The effect of different molar ratio of DES (a combination of choline chloride and propylene glycol) and DES viscosities on flavonoid yields.

the extraction efficiency. A previous study conducted by Bubalo et al. [32] also reported similar results for catechin extracted from grape skin. It should be noted that the presence of water can reduce the hydrogen bond formed between the solvent-sample, HBA-HBD, and can also affect the polarity of the DES [33,34]. A study conducted by Hammond et al. [35] proposed a transition mechanism from hydrated DES (DES with lower water content) to a DES aqueous solution (higher water content). A small amount of water in DES slightly contributed to the hydrogen bond network. The DES intermolecular bond persists and can tolerate hydration up to a certain amount of water until reaching a condition where the DES can be dissolved by water and the system becomes an aqueous solution.

We conducted further investigations at different molar ratios of choline chloride and propylene glycol with the water content maintained at 25% and the same extraction conditions. Figure 2 shows that DES consisting of choline and propylene glycol with a molar ratio of 1 : 1 gave the highest flavonoid yield ($p <$ 0.05). Increasing the amount of choline chloride in DES offered a low efficiency of extraction owing to its higher viscosity. A similar result was found in a previous study conducted by Ozturk et al. [36], where increasing the choline chloride molar ratio caused an increase in viscosity so that the total phenolic compound yield decreased. Furthermore, our results showed that increasing propylene glycol, caused a decrease in flavonoid compound yields. Along with an increase in propylene glycol and decreasing of choline chloride, the formation of hydrogen bonds decreases; meanwhile, the extraction efficiency is strongly influenced by the formation of hydrogen bonds [37]. Thus, we decided to keep using a molar ratio of 1 : 1 of choline chloride and propylene glycol for subsequent extraction and analysis procedures.

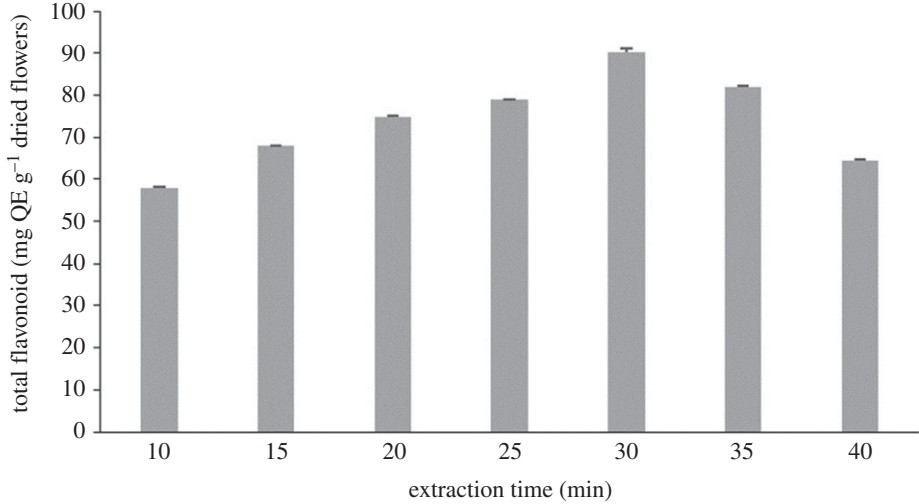

**Figure 3.** The effect of different extraction times on flavonoid yields.

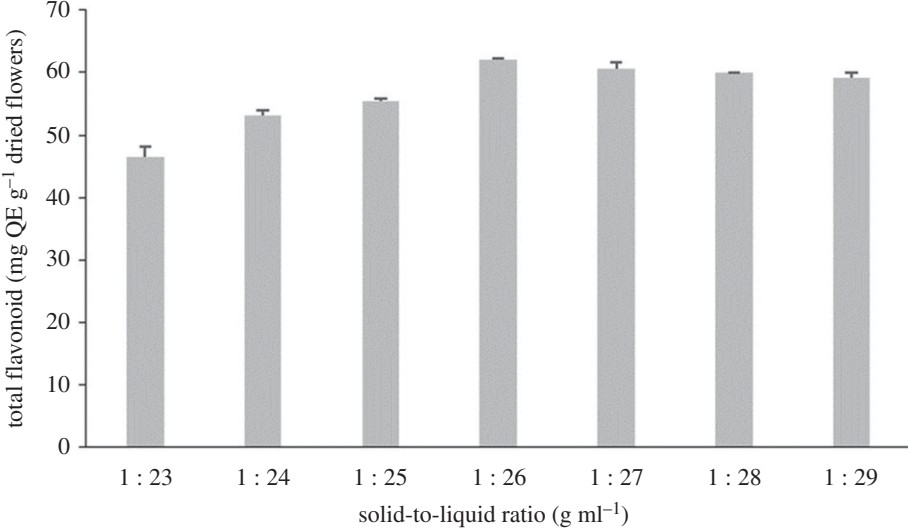

**Figure 4.** The effect of different solid-to-liquid ratios on flavonoid yields.

### 3.1.2. Extraction time

In the UAE method, the acoustic cavitation phenomenon has an important role in increasing the extraction efficiency and reducing the extraction time. As shown in figure 3, the extraction time affected the extracted yields significantly. A longer contact time between the solvent and sample provided a greater opportunity for mass transfer [38]. Our findings showed that the best extraction time was 30 min ($p < 0.05$) with the total flavonoid yields of 90.193 mg QE g$^{-1}$ dried flowers. Surprisingly, flavonoid compound yields declined with a greater increase in extraction time after 30 min. A similar trend was found in several previous studies reported by Syakfanaya *et al.* [39]. A study conducted by Suhaimi *et al.* [40] reported that UAE of more than 20 min caused a decrease of phenolic compound yields.

### 3.1.3. Solid-to-liquid ratio

The solid-to-liquid ratio is the ratio between the mass of solid plant materials to the volume of solvents. When the solid-to-liquid ratio is lower, the volume of solvents used for extraction is higher, thus increasing the dissolving ability of the compound. Several studies determined the suitable solid-to-liquid ratio for flavonoid extraction [41–43]. Figure 4 shows that along with increasing the solvent used, the yields of flavonoid compounds also greatly increased ($p < 0.05$) until reaching a maximum

**Table 2.** Experimental responses for different combination of variables.

| run | independent variable | | | response |
| | $X_1$ | $X_2$ | $X_3$ | total flavonoid (mg QE g$^{-1}$ dried flowers) |
|---|---|---|---|---|
| 1 | −1 | −1 | 0 | 35.292 |
| 2 | 1 | −1 | 0 | 45.868 |
| 3 | 0 | −1 | −1 | 38.025 |
| 4 | −1 | 1 | 0 | 16.389 |
| 5 | −1 | 0 | 1 | 61.339 |
| 6 | −1 | 0 | −1 | 65.156 |
| 7 | 0 | 1 | −1 | 38.075 |
| 8 | 1 | 1 | 0 | 44.57 |
| 9 | 0 | 0 | 0 | 63.527 |
| 10 | 0 | 0 | 0 | 63.908 |
| 11 | 0 | −1 | 1 | 36.474 |
| 12 | 1 | 0 | 1 | 72.108 |
| 13 | 0 | 1 | 1 | 17.061 |
| 14 | 1 | 0 | −1 | 89.732 |
| 15 | 0 | 0 | 0 | 63.488 |

of 61.995 mg QE g$^{-1}$ dried flowers at a solid-to-liquid ratio of 1 : 26 g ml$^{-1}$ and then slightly decreased or tended to be constant. This may indicate the appropriate volume of solvent needed to dissolve the flavonoid compounds in the materials. A similar result was also shown in previous work conducted by Jing *et al*. [3]. When the volume of solvent exceeded the appropriate volume, more impurities were also dissolved out and hindered the flavonoid dissolution.

## 3.2. Optimum extraction conditions of flavonoids

A Box Behnken design for the response surface methodology was used to optimize the extraction of flavonoid compounds. About 15 runs of the RSM experiment were performed to verify the predictive model. The extraction variables optimized in this study were the extraction time, water content and solid-to-liquid ratio (table 1). The variables and responses observed in all experimental runs are shown in table 2. All the experimental data were analysed using software and generated a regression equation as follows:

$$Y = 63.64 + 9.26x_1 - 4.95x_2 - 5.50x_3 + 4.40x_1x_2 - 3.45x_1x_3 - 4.87x_2x_3 + 5.78x_1^2 - 33.89x_2^2 + 2.66x_3^2,$$

where $Y$ represents the total flavonoid yields; $x_1$ represents the extraction time, $x_2$ represents the water content in DES; and $x_3$ represents the solid-to-liquid ratio.

The mathematical equation was used as a model to determine the relationship between the variables and response. The $R^2$ value shows that the model can express variances of more than 99.54% ($R^2$ = 0.9954). Figure 5 also demonstrates that there was considerable agreement between the actual experimental results and the predicted value of total flavonoid yields. Furthermore, an analysis using ANOVA was conducted to evaluate the quality of the model. Table 3 shows that the lack-of-fit was not significant, where $p = 0.080$ (greater than 0.05). The lack-of-fit value of a good model must be insignificant. This means that the failure of the model to represent the data is not significant, so the model is appropriate to predict the responses [40]. A $p$-value < 0.05 in the model indicates a significant effect of the variable on the response and our results showed that there are interactions between the variables.

The response surface plot on three-dimensional surface graphs can be seen in figure 6. To investigate the interactive effect of variables on total flavonoid yields, a variable was held at zero level, while varying other two variables. As expected, it can be seen from figure 6a and c that a quadratic effect of water

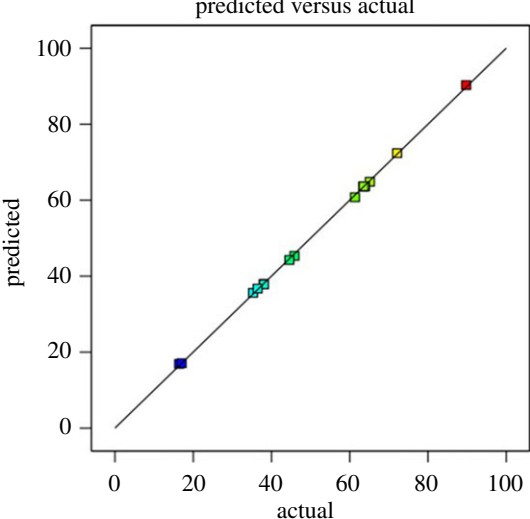

**Figure 5.** Correlation graph of predicted values and actual yields of total flavonoid compounds.

**Table 3.** Analysis of variance for the prediction model of flavonoid yields.

| source | sum of squares | degrees of freedom | mean square | F-value | p-value |
|---|---|---|---|---|---|
| model | 5928.25 | 9 | 658.69 | 1820.99 | <0.0001 |
| $X_1$ | 686.39 | 1 | 686.39 | 1897.55 | <0.0001 |
| $X_2$ | 195.66 | 1 | 195.66 | 540.92 | <0.0001 |
| $X_3$ | 242.07 | 1 | 242.07 | 669.20 | <0.0001 |
| $X_1X_2$ | 77.48 | 1 | 77.48 | 214.21 | <0.0001 |
| $X_1X_3$ | 47.66 | 1 | 47.66 | 131.75 | <0.0001 |
| $X_2X_3$ | 94.70 | 1 | 94.70 | 261.81 | <0.0001 |
| $X_1^2$ | 123.43 | 1 | 123.43 | 341.24 | <0.0001 |
| $X_2^2$ | 4241.52 | 1 | 4241.52 | 11725.87 | <0.0001 |
| $X_3^2$ | 26.14 | 1 | 26.14 | 72.27 | 0.0004 |
| residual | 1.81 | 5 | 0.3617 | | |
| lack-of-fit | 1.70 | 3 | 0.5670 | 10.53 | 0.0880 |
| pure error | 0.1077 | 2 | 0.0538 | | |
| cor total | 5930.06 | 14 | | | |

content was detected for total flavonoid yields. The extraction yields of flavonoid reached a maximum level at water content in DES of 25%. From the surface plot (figure 6*b* and *c*), the extraction time exerted an increasing trend on the response variable. The longer extraction time means an increasing of solvent contact with the sample and allows more diffusion of compound. Otherwise, the solid to liquid ratio showed a decreasing trend when the amount of solvent was increased.

Experimentally, the maximum flavonoid compound yield was 89.732 mg QE $g^{-1}$ dried flowers at an extraction time of 40 min, with 25% water content in DES, and a solid-to-liquid ratio of 1 : 25 g $ml^{-1}$. The experimental results were very close to the predicted value of 90.299 mg QE $g^{-1}$ dried flowers.

## 3.3. Comparison with a conventional extraction solvent

Ethanol is used as a comparison because it is an organic solvent that widely used as an extracting solvent. To investigate the efficiency of DES as an alternative solvent for the extraction of flavonoids from the *I. javanica* flower, the extract obtained under optimum conditions (§4.2) was compared with the

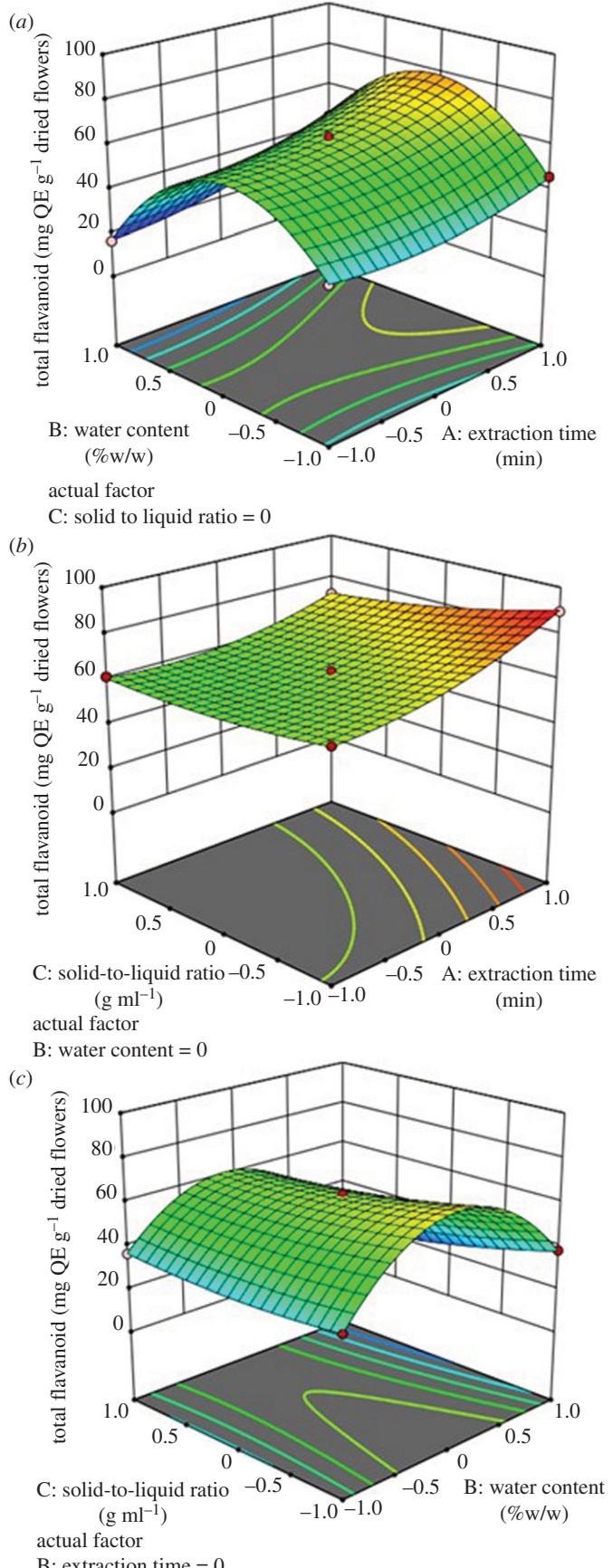

**Figure 6.** Three-dimensional response surface graphs of (a) the flavonoid yield versus extraction time ($x1$) and water content ($x2$); (b) flavonoid yield versus extraction time ($x_1$) and solid-to-liquid ratio ($x_3$); (c) flavonoid yield versus water content ($x_2$) and solid-to-liquid ratio ($x_3$).

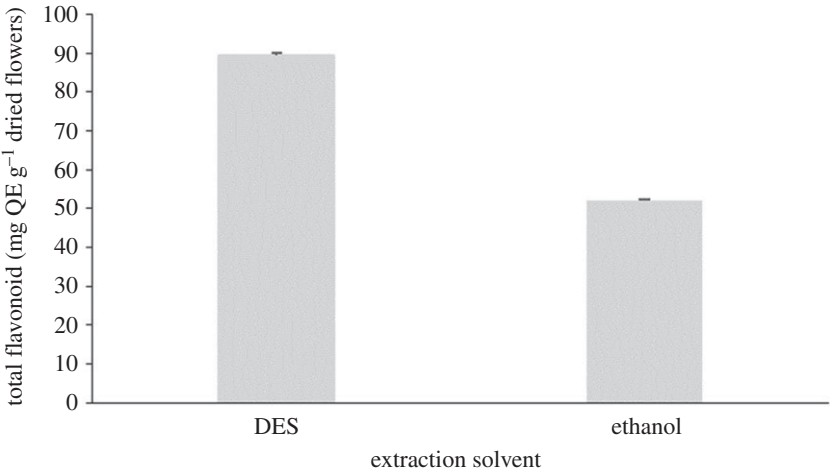

**Figure 7.** Comparison of the total flavonoid yields from DES and ethanol.

**Figure 8.** SEM image at a magnification of 500x of *I. javanica* flower powder (*a*) before the extraction process and after extraction using (*b*) ethanol, (*c*) ChCl-Pg (1 : 1), (*d*) ChCl-Pg (2 : 1), and (*e*) ChCl-Pg (1 : 2).

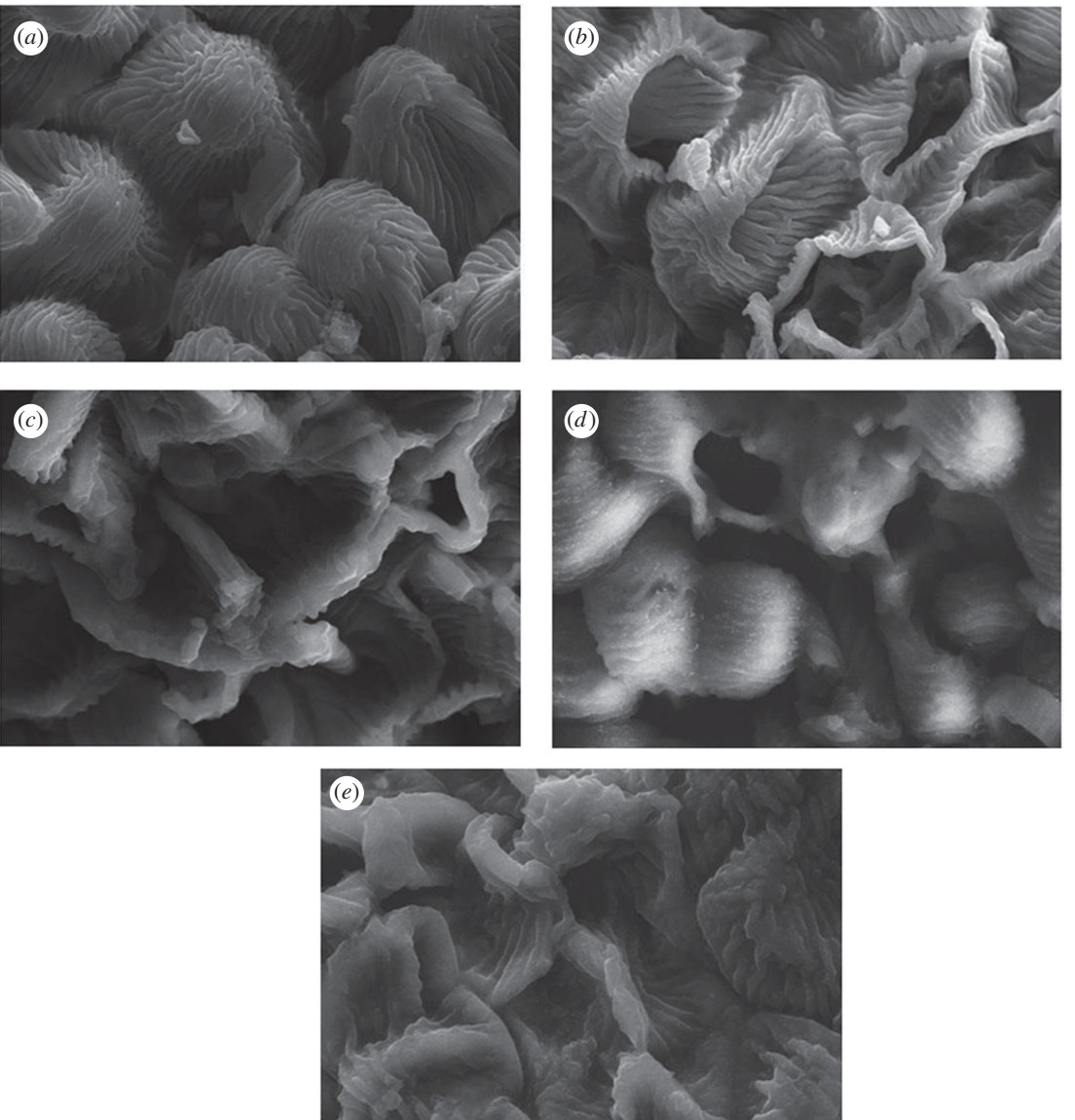

**Figure 9.** SEM image at a magnification of 5000x of *I. javanica* flower powder (*a*) before extraction process and after extraction using (*b*) ethanol, (*c*) ChCl-Pg (1 : 1), (*d*) ChCl-Pg (2 : 1), and (*e*) ChCl-Pg (1 : 2).

ethanolic extract. Under the same conditions, ethanol, as a conventional organic solvent, provided significantly lower total flavonoid compound quantities than DES (shown in figure 7). Thus, it can be concluded that DES has better extraction efficiency than ethanol.

## 3.4. Scanning electron microscope analysis

The outer surface of the *I. javanica* flower after the extraction process was captured and compared with the outer surface before extraction (shown in figures 8 and 9). Figures 8*a* and 9*a* show that the surface of the *I. javanica* flower particle was compact and undamaged. This structure changed after extraction using the UAE method. Many studies reported that ultrasound cavitation impacts the surface of solid plant materials, which subsequently becomes the mechanism leading to extraction enhancements in UAE, such as fragmentation, erosion, the sonocapillary effect, detexturation, local shear stress and sonoporation [44,45].

Figures 8*c,e*, and 9*c,e* illustrate that extraction using DES produced good damage on the surface of the *I. javanica* flower material, compared to that using ethanol (figures 8*b* and 9*b*). Increasing the damage intensity leads to permeation of the solvent and facilitated diffusion process and mass transfer.

However, figures 8*d* and 9*d* show that there was only slight cell wall damage. This was probably affected by the viscosity of DES used for UAE. As mentioned previously, because of the higher choline chloride content in DES, flavonoid yields were decreased owing to the higher viscosity. In a high viscosity environment, the microbubble formation in the cavitation phenomenon will be disrupted, thereby decreasing the extraction efficiency [46,47].

## 4. Conclusion

A combination of the UAE method and DES is a promising alternative environmentally friendly extraction method for obtaining high flavonoid yields from the *I. javanica* flower. Different molar ratio and viscosity values of DES, as well as water content, extraction time and solid-to-liquid ratio values influenced the total flavonoid yields. The use of DES as an extraction solvent not only decreased the side effects of the toxic organic solvent but also have proven to offer better capabilities in plant bioactive compounds compared to conventional solvent extraction.

Data accessibility. The data supporting the results in this article can be accessed at the Dryad Digital Repository: https://dx.doi.org/10.5061/dryad.fbg79cns9 [48].

Competing interests. The authors declare no competing interest.

Funding. The authors appreciate the financial support provided by The Ministry of Research, Technology and Higher Education of the Republic of Indonesia for PDUPT 2020 grant ref no. NKB-86/UN2.RST/HKP.05.00/2020. N.D.O. was also supported by the Universitas Surabaya, Indonesia, with a PhD programme scholarship.

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
