## [Reviewer comments · Royal Society Open Science]

Review History

RSOS-201116.R0 (Original submission)

Review form: Reviewer 1

Is the manuscript scientifically sound in its present form?

Yes

Are the interpretations and conclusions justified by the results?

Yes

Is the language acceptable?

Yes

Do you have any ethical concerns with this paper?

No

Have you any concerns about statistical analyses in this paper?

Yes

Recommendation?

Major revision is needed (please make suggestions in comments)

Comments to the Author(s)

Title: flowers instead of flower.

Abstract:

Can be formulated shorter without losing information. Add why flavonoids in *I. javanica* are so important.

Introduction:

First sentence: biological activities in plants? This is followed by biological activities in humans. Transfer information from the one but last paragraph to the Discussion and align with the text there. It contains too much details for an introduction.

Results and discussion:

Check legends of figures and tables. They are often confusing, lack relevant information. Table 3: x-axis molar ratio of what?

I miss statistical evaluation of extraction time.

4.2: refer to Table 1 as well.

Section 4.2 should be rewritten and reorganized and better aligned with the Methods section.

Explain the BB design to the reader.

Figure 6: add units.

A long extraction results loss of flavonoid content. What happens? You measure the total content calculated as quercetin. Add temperature measurements of the extraction mixture as a function of time.

Was extraction exhaustive?

In 4.4 the authors talk about 'porosity'. Please define this in the context of your work.

The authors present the extraction as 'green'. Of course, green is hot and sells, but work this out a bit better in the discussion. An important aspect, not mentioned by you, is how to get rid of the solvent. Ethanol easily evaporated, but the mixtures you used? How are the extracts applied in herbal medicine? This would place the work in a broader perspective. What do you want to achieve?

Figure 7 looks nice, but is hardly explained in the text. Please do so. Take the reader by the hand and lead him/her along your results. This can be done better!

The conclusion is rather a repetition of the summary than a real conclusion. Make it shorter and more to the point.

I recommend to reduce the number of references. A total of 55 is a lot for a study like this.

Confine to the most relevant ones, it is not a review.

Further, I have placed comment balloons in the manuscript with comments and specific suggestions for some textual and stylistic improvements. This is attached (Appendix A). Consider those carefully. No problem that my name is shown to the authors.

In conclusion: interesting work that merits publication. However, by presenting it in a better way it can really be improved.

Review form: Reviewer 2**Is the manuscript scientifically sound in its present form?**

Yes

Are the interpretations and conclusions justified by the results?

Yes

Is the language acceptable?

Yes

Do you have any ethical concerns with this paper?

No

Have you any concerns about statistical analyses in this paper?

No

Recommendation?

Accept with minor revision (please list in comments)

Comments to the Author(s)

The manuscript "A Green Extraction Design for Enhancing Flavonoid Compounds from the *Ixora javanica* flower using a Deep Eutectic Solvent" reported by Mun'im et al. is a very interesting work to obtain flavonoid compounds by a combination of ultrasound-assisted (UAE) extraction method and deep eutectic solvent (DES) under room temperature. The authors discussed the influences of many extraction parameters on flavonoid yields, including different molar ratio of DES, viscosity of DES, water content, extraction time, and solid-to-liquid ratio. Response surface analyses using three-level and three-factor Box Behnken designs were also performed to give the optimum flavonoid concentrations. Overall, the experiments were well designed, the results were clear. I would suggest to publish it in Royal Society Open Science after minor revision.

1. Many formats about abbreviations and figures are not uniform in the manuscript. It should be carefully checked and revised.

For instance, "*I. javanica*" (Page 2, lines 42 and 43) should be italic;

"P" (Page 5, line 3) represents the abbreviation of propylene glycol, it look not proper;

The ordinate scale in Figures 1, 4, 5 should be given as displayed in Figures 2 and 3.

2. The detailed reaction conditions except for corresponding variable in Figures 2, 3, 4, 5 should be given.

3. In the section of 4.1 (Page 4, line 48), the authors discussed the variation trend of flavonoid yields with change of different extraction variables, but the detailed values of flavonoid yields are not mentioned.

4. In order to make reader better understand the meaning of mathematical equation of the model (Page 7, line 10), its source should be given.

5. This paper described the extraction efficiency of flavonoid from *Ixora javanica* flower using DES is superior to conventional extraction (ethanol), indicating DES is an effective method for flavonoid extraction. As we know, ethanol solvent is easy to recycle for next utilization, so the authors should consider the recyclability of DES in this manuscript.

Decision letter (RSOS-201116.R0)

Dear Miss Oktaviyanti:

Title: A Green Extraction Design for Enhancing Flavonoid Compounds from the *Ixora javanica* flower using a Deep Eutectic Solvent

Manuscript ID: RSOS-201116

The editor assigned to your manuscript has now received comments from reviewers. We would like you to revise your paper in accordance with the referee and Subject Editor suggestions which can be found below (not including confidential reports to the Editor). Please note this decision does not guarantee eventual acceptance.

Please submit your revised paper before 22-Aug-2020. Please note that the revision deadline will expire at 00.00am on this date. If we do not hear from you within this time then it will be assumed that the paper has been withdrawn. In exceptional circumstances, extensions may be possible if agreed with the Editorial Office in advance. We do not allow multiple rounds of revision so we urge you to make every effort to fully address all of the comments at this stage. If deemed necessary by the Editors, your manuscript will be sent back to one or more of the original reviewers for assessment. If the original reviewers are not available we may invite new reviewers.

On behalf of the Subject Editor Professor Anthony Stace and the Associate Editor Dr Darren Walsh.

RSC Associate Editor:
Comments to the Author:
(There are no comments.)

RSC Subject Editor:
Comments to the Author:
(There are no comments.)

Reviewers' Comments to Author:

Reviewer: 1

Comments to the Author(s)

Title: flowers instead of flower.

Abstract:

Can be formulated shorter without losing information. Add why flavonoids in *I. javanica* are so important.

Introduction:

First sentence: biological activities in plants? This is followed by biological activities in humans. Transfer information from the one but last paragraph to the Discussion and align with the text there. It contains too much details for an introduction.

Results and discussion:

Check legends of figures and tables. They are often confusing, lack relevant information. Table 3: x-axis molar ratio of what?

I miss statistical evaluation of extraction time.

4.2: refer to Table 1 as well.

Section 4.2 should be rewritten and reorganized and better aligned with the Methods section.

Explain the BB design to the reader.

Figure 6: add units.

A long extraction results loss of flavonoid content. What happens? You measure the total content calculated as quercetin. Add temperature measurements of the extraction mixture as a function of time.

Was extraction exhaustive?

In 4.4 the authors talk about 'porosity'. Please define this in the context of your work.

The authors present the extraction as 'green'. Of course, green is hot and sells, but work this out a bit better in the discussion. An important aspect, not mentioned by you, is how to get rid of the solvent. Ethanol easily evaporated, but the mixtures you used? How are the extracts applied in herbal medicine? This would place the work in a broader perspective. What do you want to achieve?

Figure 7 looks nice, but is hardly explained in the text. Please do so. Take the reader by the hand and lead him/her along your results. This can be done better!

The conclusion is rather a repetition of the summary than a real conclusion. Make it shorter and more to the point.

I recommend to reduce the number of references. A total of 55 is a lot for a study like this.

Confine to the most relevant ones, it is not a review.

Further, I have placed comment balloons in the manuscript with comments and specific suggestions for some textual and stylistic improvements. This is attached. Consider those carefully. No problem that my name is shown to the authors.

In conclusion: interesting work that merits publication. However, by presenting it in a better way it can really be improved.

Reviewer: 2

Comments to the Author(s)

The manuscript "A Green Extraction Design for Enhancing Flavonoid Compounds from the *Ixora javanica* flower using a Deep Eutectic Solvent" reported by Mun'im et al. is a very interesting work to obtain flavonoid compounds by a combination of ultrasound-assisted (UAE) extraction method and deep eutectic solvent (DES) under room temperature. The authors discussed the

influences of many extraction parameters on flavonoid yields, including different molar ratio of DES, viscosity of DES, water content, extraction time, and solid-to-liquid ratio. Response surface analyses using three-level and three-factor Box Behnken designs were also performed to give the optimum flavonoid concentrations. Overall, the experiments were well designed, the results were clear. I would suggest to publish it in Royal Society Open Science after minor revision.

1. Many formats about abbreviations and figures are not uniform in the manuscript. It should be carefully checked and revised.

For instance, "*I. javanica*" (Page 2, lines 42 and 43) should be italic;

"P" (Page 5, line 3) represents the abbreviation of propylene glycol, it look not proper;

The ordinate scale in Figures 1, 4, 5 should be given as displayed in Figures 2 and 3.

2. The detailed reaction conditions except for corresponding variable in Figures 2, 3, 4, 5 should be given.

3. In the section of 4.1 (Page 4, line 48), the authors discussed the variation trend of flavonoid yields with change of different extraction variables, but the detailed values of flavonoid yields are not mentioned.

4. In order to make reader better understand the meaning of mathematical equation of the model (Page 7, line 10), its source should be given.

5. This paper described the extraction efficiency of flavonoid from *Ixora javanica* flower using DES is superior to conventional extraction (ethanol), indicating DES is an effective method for flavonoid extraction. As we know, ethanol solvent is easy to recycle for next utilization, so the authors should consider the recyclability of DES in this manuscript.

Author's Response to Decision Letter for (RSOS-201116.R0)

See Appendix B.

RSOS-201116.R1 (Revision)

Review form: Reviewer 1

Is the manuscript scientifically sound in its present form?

Yes

Are the interpretations and conclusions justified by the results?

Yes

Is the language acceptable?

Yes

Do you have any ethical concerns with this paper?

No

Have you any concerns about statistical analyses in this paper?

No

Recommendation?

Accept as is

Comments to the Author(s)

In my opinion this revised manuscript is acceptable for publication. The referees' comments have been well elaborated. Overall, the manuscript has improved significantly.

Review form: Reviewer 2

Is the manuscript scientifically sound in its present form?

Yes

Are the interpretations and conclusions justified by the results?

Yes

Is the language acceptable?

Yes

Do you have any ethical concerns with this paper?

No

Have you any concerns about statistical analyses in this paper?

No

Recommendation?

Accept as is

Comments to the Author(s)

The author had revised current manuscript according to my comments, and I am satisfied with these responses. Therefore, I recommend this manuscript to publish in Royal Society Open Science in its current state.

Decision letter (RSOS-201116.R1)

Dear Miss Oktavianti:

Title: A Green Extraction Design for Enhancing Flavonoid Compounds from the *Ixora javanica* flowers using a Deep Eutectic Solvent

Manuscript ID: RSOS-201116.R1

It is a pleasure to accept your manuscript in its current form for publication in Royal Society Open Science. The chemistry content of Royal Society Open Science is published in collaboration with the Royal Society of Chemistry.

On behalf of the Subject Editor Professor Anthony Stace and the Associate Editor Dr Darren Walsh.

RSC Associate Editor:
Comments to the Author:
(There are no comments.)

RSC Subject Editor:
Comments to the Author:
(There are no comments.)

Reviewer(s)' Comments to Author:
Reviewer: 1

Comments to the Author(s)
In my opinion this revised manuscript is acceptable for publication. The referees' comments have been well elaborated. Overall, the manuscript has improved significantly.

Reviewer: 2

Comments to the Author(s)
The author had revised current manuscript according to my comments, and I am satisfied with these responses. Therefore, I recommend this manuscript to publish in Royal Society Open Science in its current state.

Appendix A**ROYAL SOCIETY
OPEN SCIENCE****A Green Extraction Design for Enhancing Flavonoid
Compounds from the *Ixora javanica* flower using a Deep
Eutectic Solvent**

Journal:	Royal Society Open Science
Manuscript ID	RSOS-201116
Article Type:	Research
Date Submitted by the Author:	25-Jun-2020
Complete List of Authors:	Oktaviyanti, Nina Dewi; University of Surabaya, Department of Pharmaceutical Biology, Faculty of Pharmacy; University of Indonesia, Department of Pharmacognosy-Phytochemistry, Faculty of Pharmacy Kartini, Kartini; University of Surabaya, Department of Pharmaceutical Biology, Faculty of Pharmacy Hadiyat, Mochammad Arbi; University of Surabaya, Department of Industrial Engineering, Faculty of Engineering Rachmawati, Ellen; University of Surabaya, Department of Pharmaceutical Biology, Faculty of Pharmacy Wijaya, Andre Chandra; University of Surabaya, Department of Pharmaceutical Biology, Faculty of Pharmacy Hayun, Hayun; Universitas Indonesia, Graduate Program of Herbal Medicine, Faculty of Pharmacy Mun'im, Abdul; Universitas Indonesia, Department of Pharmacognosy-Phytochemistry, Faculty of Pharmacy; Universitas Indonesia, Graduate Program of Herbal Medicine, Faculty of Pharmacy
Subject:	Green chemistry < CHEMISTRY, plant science < BIOLOGY, Analytical chemistry < CHEMISTRY
Keywords:	Ixora javanica , flavonoid, Response surface methodology, deep eutectic solvent, ultrasound assisted extraction
Subject Category:	Chemistry

Author-supplied statements

Relevant information will appear here if provided.

Ethics

Does your article include research that required ethical approval or permits?:

This article does not present research with ethical considerations

Statement (if applicable):

CUST_IF_YES_ETHICS :No data available.

Data

It is a condition of publication that data, code and materials supporting your paper are made publicly available. Does your paper present new data?:

Yes

Statement (if applicable):

The data supporting the results in this article can be accessed at the Dryad Digital Repository:

<https://doi.org/10.5061/dryad.fbg79cns9>

URL for reviewers:

https://datadryad.org/stash/share/LJB2GRF16vAZ7DKK4YMSRDeH_FZh9H44OXN_GTC3J1E

Conflict of interest

I/We declare we have no competing interests

Statement (if applicable):

CUST_STATE_CONFLICT :No data available.

Authors' contributions

This paper has multiple authors and our individual contributions were as below

Statement (if applicable):

Nina Dewi Oktaviyanti: Conceived, designed, and performed the experiments; analyzed the data; wrote the paper

Kartini Kartini: Designed and supervised the study; performed the experiments; Interpreted the data

Mochammad Arbi Hadiyat: Performed the experiments; Interpreted the data and performed statistic Analysis

Ellen Rachmawati: Performed the experiments; collected the data

Andre Chandra Wijaya: Performed the experiments; collected the data

Hayun Hayun: Designed and supervised the study; performed the experiments; Interpreted the data

Abdul Mun'im: Conceived, designed, supervised the study and performed the experiments; analyzed the data; contributed reagents, materials and analysis tools.

All author contributed in paper writing.

A Green Extraction Design for Enhancing Flavonoid Compounds from the *Ixora javanica* flower using a Deep Eutectic Solvent

Nina Dewi Oktaviyanti^{1,2}, Kartini Kartini², Mochammad Arbi Hadiyat³, Ellen Rachmawati², Andre Chandra Wijaya², Hayun Hayun⁴ and Abdul Mun'im^{1,4}

¹ Department of Pharmacognosy-Phytochemistry, Faculty of Pharmacy, Universitas Indonesia, Cluster of Health Sciences Building, Depok 16424 West Java, Indonesia

² Department of Pharmaceutical Biology, Faculty of Pharmacy, University of Surabaya, Surabaya 60293, East Java, Indonesia

³ Department of Industrial Engineering, Faculty of Engineering, Universitas of Surabaya, Surabaya 60293, East Java, Indonesia

⁴ Graduate Program of Herbal Medicine, Faculty of Pharmacy, Universitas Indonesia, Kampus UI Depok 16424 West Java, Indonesia

Keywords: *Ixora javanica*; flavonoid; Response surface methodology; deep eutectic solvent; ultrasound assisted extraction

1. Summary

In this study, the development of environmentally friendly extraction methods for *Ixora javanica*, a flowering plant belonging to the family Rubiaceae, was carried out. The objectives of the present work were to provide recommendations for the optimal extraction conditions and to investigate the effects of any extraction parameters on flavonoid yields from the *I. javanica* flower. The extraction process was performed using deep eutectic solvent (DES) (choline chloride and propylene glycol at molar ratio of 1:1) and ultrasound-assisted (UAE) extraction methods. Both single-factor and response surface analyses using three-level and three-factor Box Behnken designs were conducted to obtain the optimum flavonoid concentrations. The results showed that the different molar ratio and viscosity of DES, water content, extraction time, and solid-to-liquid ratio affected the total flavonoid yields significantly. The optimum extraction conditions for total flavonoids featured an extraction time of 40 min, 25% water content in DES, and a solid-to-liquid ratio of 1:25 g/mL. The extract obtained under optimum extraction conditions showed higher total flavonoid yields than the ethanolic extract. The scanning electron microscope (SEM) images demonstrated that both of the solvents also showed different effects on the outer surface of the *I. javanica* flower during extraction process. In sum, our work succeeded in determining the optimum conditions for total flavonoids in the *I. javanica* flower using a green extraction method.

2. Introduction

Flavonoids are the most abundant secondary metabolites and are known to be responsible for various biological activities in plants. At present, flavonoids have become a concern because of their benefits against many degenerative diseases, such as cardiovascular disease, cancer, and diabetes. Many researchers have correlated the various activities of flavonoids with their very strong antioxidant activity [1-5]. Therefore, plants with a high level of flavonoids are prospective for development as herbal preparations to treat degenerative and age-related disease.

Various attempts have been made to obtain optimum flavonoid compounds from plants. Non-conventional extraction methods, such as ultrasound-assisted extraction (UAE), are often applied to improve extraction efficiency and increase bioactivity. In addition, these methods have many advantages, including environmental friendliness, the need for minimal solvent, and time efficiency [6-9]. The development of flavonoids and other plant metabolite extraction processes is not only limited to the use of organic solvents. Many previous studies have applied deep eutectic solvents (DESs) as alternative environmentally friendly solvents for the extraction of flavonoids [10-14]. Generally, DESs have been widely used to replace organic solvents and have proven to offer better capabilities in plant bioactive compounds than organic solvents. In brief, DESs are prepared from at least two components (e.g., Hydrogen Bond Acceptor (HBA) and Hydrogen Bond Donor (HBD)) at an appropriate molar ratio. The use of DESs as a green solvent in the

*Abdul Mun'im (munim@farmasi.ui.ac.id).

†Present address: Department of Pharmacognosy-Phytochemistry, Faculty of Pharmacy, Universitas Indonesia, Cluster of Health Sciences Building, Depok 16424 West Java, Indonesia

extraction process is currently very prospective because DESs are very simple, cheap, less toxic, have easy preparation, and can be adjusted according to the purpose of extraction [15-17].

I. javanica, an attractive red flowering plant that is a member of the Rubiaceae family, is known to contain high levels of flavonoid compounds, especially in its flowers. Phytochemical studies of *I. javanica* reported the presence of flavonoids such as quercetin, formononetin, and anthocyanin [18-19]. Besides flavonoid content, the flowers were also reported to contain many phenolic and terpenoid compounds, which are responsible for many activities, such as antioxidant, antitumor, anti-inflammatory, hepatoprotective, and tyrosinase inhibiting activities [20-23].

To date, studies related to the application of DES and UAE for *I. javanica* flower extraction are very limited. In our previous study, we were able to optimize the extraction variables, including extraction time, solid-to-liquid ratio, and temperature, to obtain the maximum flavonoid compounds using the response surface method [24]. In the extraction process, the viscosity of DES is a common problem [25]. The addition of water to DES mixtures can be used to solve the viscosity problems of DES [15,17,26,27]. There are no studies reporting the optimum level of water in DES needed to increase the effectivity of DES in flavonoid compound extraction from *I. javanica* flowers. Therefore, in this present work, we optimized the extraction conditions by using water content in DES as one of the extraction variables, followed by extraction time and the solid-to-liquid ratio. Unlike the previous study, here we performed the extraction at room temperature. Theoretically, the desirable extraction temperature using DES ranges from room temperature to about 60 °C [28]. The extraction temperature can increase the extraction rate by decreasing the viscosity [29]. However, if the temperature keeps increasing, the yields of the target substance that are thermally sensitive may be decreased. Moreover, the use of a high temperature in extraction may increase the energy consumption, which is contradictive to the green extraction principle [28]. A study conducted by Dai et al. [30] highlighted that the extraction of flavonoid compounds can be carried out at ambient temperatures and provided the maximum extraction recovery for flavonoid compounds.

The main purpose of this study is not only to provide recommendations for the optimum extraction conditions using the response surface method but also to investigate the effect of each extraction variable on flavonoid compound yields by using single-factor experiments. A single-factor analysis was first employed in a preliminary study before optimization using the response surface method. To offer a better explanation of extraction efficiency, we also performed SEM imaging on dried flower powder.

3. Materials and Methods

3.1. Chemicals and materials

Choline chloride was purchased from Xi'an Rongsheng Biotechnology Co, Ltd, China, while other solvents such as propylene glycol and ethanol, were acquired from Merck, Germany. The quercetin standards used in this study were purchased from Sigma Aldrich, USA.

3.2. Plant materials

The fresh red color *I. javanica* flower used in this study was authenticated by the Center for Traditional Medicine Information and Development, Faculty of Pharmacy, University of Surabaya. The flower was collected at full bloom from the Tenggilis Mejoyo district, Surabaya, East Java, Indonesia. After sortation, the flowers were washed, shade dried, and powdered. The dry powder was then sieved through a size 30 mesh sifter, stored in a sealed container, and kept under dry environmental conditions.

3.3. Preparation of DES

The DES components used in this study were choline chloride (ChCl) as HBA and propylene glycol (Pg) as HBD. DESs were prepared by heating and stirring both components at certain molar ratios (1:1; 1:2; 2:1) in a flask at 50 °C for 30 min constantly until a homogeneous clear mixture formed. Deionized water was added to the DES to prepare the DES solution containing various concentrations of water. The water content used in this study was 5%–35% w/w of water in the DES. All DES mixtures obtained in this study were stable in liquid form under room temperature storage. The DES mixtures were measured for their viscosity before being used for the extraction process (Brookfields™ cone plate viscometer, Ametek Brookfield, Middleboro, USA).

3.4. Extraction Procedures

The extraction of *I. javanica* flowers was carried out using the UAE method. Dried flower powder was weighed and extracted using DES that was prepared at a certain solid-to-liquid ratio and then sonicated for many repetitions at room temperature. When investigating the effect of one variable on the flavonoid yields, the other variables were kept constant. To determine the effect of water content in DES on the flavonoid yields, 0.5 g dried powder was extracted with 10 mL DES (a combination of ChCl-Pg at a molar ratio of 1:1), which contained different concentrations of water in DES (5%, 10%, 15%, 20%, 25%, 30%, and 35% w/w) for 5 min. Extraction time: 0.5 g dried powder was mixed with 10 mL DES (a combination of ChCl-Pg at a molar ratio of 1:1 and water content of 20% w/w), and the extraction was carried out under different extraction times (10, 15, 20, 25, 30, 35, and 40 min). The solid-to-liquid ratio: 0.5 g dried powder was mixed with different volumes of DES (a combination of ChCl-Pg at a molar ratio of 1:1 and water content of 20% w/w) until reaching a certain solid-to-liquid ratio (1:23, 1:24, 1:25, 1:26, 1:27, 1:28, and 1:29 g/ml) and was then sonicated for 5 min. The ultrasonic bath was set at room temperature with a fixed frequency radiation of 40 kHz. The extracts obtained were then centrifuged at 1500 rpm for 15 min, and the filtrates were collected. The filtrate was then analyzed for its flavonoid levels. Extraction using a conventional extraction solvent was carried out by sonicating 0.5 g dried material using ethanol under the optimum solid-to-liquid ratio and extraction time obtained from the RSM analysis. The flavonoid levels in the ethanolic extract were compared with those in the DES extracts. All procedures of extraction in this study were conducted in triplicate.

3.5. Determination of total flavonoids

The total levels of flavonoids in the extracts were determined using the method applied in the previous study with very slight modifications [24]. About 1.0 mL of each extract's filtrates was mixed with 1.5 mL of 0.33% AlCl₃ and 1.5 mL of 10% acetic acid solution. The mixtures were added with 96% ethanol until reaching a total volume of 10.0 ml. After 30 min incubation, the mixed solutions were analyzed spectrophotometrically (UV-1900, Shimadzu Corp, Kyoto, Japan) at 425.8 nm. Quercetin was used as a standard compound so that the total flavonoid levels in the dried flower powder could be expressed as their quercetin equivalent (mg QE/ g dried flowers). A calibration curve and regression equation were used to calculate the flavonoid levels in the extract sample and are shown in figure 1. All of the analysis procedures were conducted in triplicate.

Figure 1. Calibration curve of the quercetin standard

3.6. Extraction optimization using RSM

To optimize the extraction process of the *I. javanica* flower to obtain the highest total flavonoid level, three independent variables, each of which consisted of three levels, were statistically analyzed by a Box Behnken design using the Response Surface Methodology (RSM). These three levels of variables were obtained from the analysis of each extraction parameter using a single factor. All variables and selected levels selected are represented in Table 1.

Table 1. The code, range, and level of each variable selected for the experimental design.

Variables	Unit	Code	Range and level (xi)		
			-1	0	1
Time	min	X ₁	20	30	40
Water content in DES	%	X ₂	15	25	35
Solid-to-liquid ratio	g/mL	X ₃	1/25	1/26	1/27

3.7. Scanning electron microscopy (SEM)

Dried flower powder, both before and after the extraction process, was coated with 5 nm of Au before the imaging process using a Scanning Electron Microscope (Inspect S50, Fei, Japan) at magnifications of 500x and 5000x. The SEM imaging was performed to visualize the surfaces of the solid materials and correlate then with the extraction efficiency.

3.8 Data Analysis

In this study, the effects of extraction parameters on flavonoid yields were analyzed via single-factor assisted one-way analysis of variance (ANOVA) test ($p < 0.05$ significance level) (SPSS software version 18 for Windows, IBM, New York, United States) and the response surface method (Design Expert Software, Version 11, Stat-Ease Inc., Minneapolis, MN, USA). All total flavonoid data in the text and tables are presented as the mean \pm standard deviation (SD); additionally, the SD is indicated as error bars in the figures.

4. Results and discussion

4.1. Effect of different extraction variables on flavonoid yields

4.1.1. Viscosity and water content in DES

The selection of the extraction solvent is crucial to increase the acquisition of the target compounds. There are many kinds of HBA and HBD that can be used for DES preparation [31,32]. A combination of HBAs and HBDs in DES preparation can affect the characteristics, polarity, and dissolving ability of DESs [33-35]. The selection of HBA and HBD should be adjusted to the target compound according to the "like dissolve like" principle. Our previous study reported that a combination of choline chloride (ChCl) as an HBA and propylene glycol (Pg) as an HBD offers the most effective flavonoid extraction from *I. javanica* flowers out of 11 other types of DES [24]. As mentioned previously, water can be added to DES to solve its high viscosity problem. A high viscosity extraction solvent makes mass transfer difficult and decreases the extraction yields [36,37]. Our results demonstrated that water addition resulted in a lower viscosity of DES. Figure 2 shows that adding a greater amount of water into DES resulted in higher extraction yields, which reached the maximum yield at a concentration of 25%. Furthermore, the addition of water above 25% subsequently decreased the extraction efficiency. A previous study conducted by Bubalo et al. [38] also reported the same results for catechin extracted from grape skin. It should be noted that the presence of water can reduce the hydrogen bond formed between the solvent-sample, HBA-HBD, and can also affect the polarity of the DES [39,40]. A study conducted by Hammond et al. [41] proposed a transition mechanism from hydrated DES (DES with lower water content) to a DES aqueous solution (higher water content). A small amount of water in DES slightly contributed to the hydrogen-bond network. The DES intermolecular bond persists and can tolerate hydration up to a certain amount of water until reaching a condition where the DES can be dissolved by water and the system becomes an aqueous solution.

We conducted further investigations at different molar ratios of ChCl-Pg with the water content maintained at 25% and the same extraction conditions. Figure 3 shows that DES consisting of ChCl and Pg with a molar ratio of 1:1 gave the highest flavonoid yield ($p < 0.05$). Increasing the amount of ChCl in DES offered a low efficiency of extraction due to its higher viscosity. A similar result was found in a previous study conducted by Ozturk et al. [42], where increasing the ChCl molar ratio caused an increase in viscosity so that the total phenolic compound yield decreased. Furthermore, our results showed that increasing P, caused a decrease in flavonoid compound yields. Along with an increase Pg and decreasing of ChCl, the formation of hydrogen bonds decreases; meanwhile, the extraction efficiency is strongly influenced by the formation of hydrogen bonds [43]. Thus, we decided to keep using a molar ratio of 1:1 of ChCl and Pg for subsequent extraction and analysis procedures.

Figure 2. The effect of different water quantities in DES on flavonoid yields

Figure 3. The effect of different molar ratios of DES (a combination of choline chloride and propylene glycol) and DES viscosities on flavonoid yields

4.1.2. Extraction time

As shown in Figure 4, the extraction time affected the extracted yields significantly. A longer contact time between the solvent and sample provided a greater opportunity for mass transfer [44]. Surprisingly, our findings showed that flavonoid compound yields declined with greater increase in extraction time after 30 min. In the UAE method, the acoustic cavitation phenomenon has an important role in increasing the extraction efficiency and reducing the extraction time. However, a longer extraction time can affect the stability of the compound since more energy is released, and the temperature also increased [3,45,46]. A study conducted by Suhaimi et al. [47] reported that ultrasound-assisted extraction of more than 20 min caused the oxidation of phenolic compounds.

Figure 4. The effect of different extraction times on flavonoid yields

4.1.3. Solid-to-liquid ratio

The solid-to-liquid ratio is the ratio between the mass of solid plant materials to the volume of solvents. When the solid-to-liquid ratio is lower, the volume of solvents used for extraction is higher, thus increasing the dissolving ability of the compound. Several studies determined the suitable solid-to-liquid ratio for flavonoid extraction [48-51]. Figure 5 shows that along with increasing the solvent used, the yields of flavonoid compounds also greatly increased until reaching their maximum at a solid-to-liquid ratio of 1:26 g/mL and then slightly decreased or tended to be constant. This may indicate the appropriate volume of solvent needed to dissolve the flavonoid compounds in the materials. A similar result was also shown in previous work conducted by Jing et al. [3]. When the volume of solvent

Figure 5. The effect of different solid-to-liquid ratios on flavonoid yields

exceeded the appropriate volume, more impurities were also dissolved out and hindered the flavonoid dissolution

4.2. Optimum extraction conditions of flavonoids

A Box Behken design for the response surface methodology was used to optimize the extraction of flavonoid compounds. About 15 runs of the RSM experiment were performed to verify the predictive model. The extraction variables optimized in this study were the extraction time, water content, and solid-to-liquid ratio. The variables and responses observed in all experimental runs are shown in Table 2. The mathematical equation of the model used to determine the relationship between the variables and response was:

$$Y = 63.64 + 9.26x_1 - 4.95x_2 - 5.50x_3 + 4.40x_1x_2 - 3.45x_1x_3 - 4.87x_2x_3 + 5.78x_1^2 - 33.89x_2^2 + 2.66x_3^2$$

where Y represents the total flavonoid yields; x_1 represents the extraction time, x_2 represents the water content in DES; and x_3 represents the solid-to-liquid ratio.

The R^2 value shows that the model can express variances of more than 99.54% ($R^2=0.9954$). Figure 6 also demonstrates that there was considerable agreement between the actual experimental results and the predicted value of total flavonoid yields. Furthermore, an analysis using ANOVA was conducted to evaluate the quality of the model. Table 3 shows that the lack-of-fit was not significant, where $p=0.080$ (>0.05). The lack-of-fit value of a good model must be insignificant. This means that the failure of the model to represent the data is not significant, so the model is appropriate to predict the responses [47]. A small p value in the model indicates a significant effect of the variable on the response. Our results showed that there are interactions between the variables (shown by $p<0.05$). The response surface plot on 3D surface graphs can be seen in Figure 7. Experimentally, the maximum flavonoid compound yield was 89.732 mg QE/ g dried flowers at an extraction time of 40 min, with 25% water content in DES, and a solid-to-liquid ratio of 1:25 g/mL. The experimental results were very close to the predicted value of 90.299 mg QE/ g dried flowers.

Table 2. Experimental responses for different combination of variables.

Run	Independent variable			Response
	X_1	X_2	X_3	Total flavonoid (mg QE/ g dried flowers)
1	-1	-1	0	35.292
2	1	-1	0	45.868
3	0	-1	-1	38.025
4	-1	1	0	16.389
5	-1	0	1	61.339
6	-1	0	-1	65.156
7	0	1	-1	38.075
8	1	1	0	44.57
9	0	0	0	63.527
10	0	0	0	63.908
11	0	-1	1	36.474
12	1	0	1	72.108
13	0	1	1	17.061
14	1	0	-1	89.732
15	0	0	0	63.488

Table 3. Analysis of variance for the prediction model of flavonoid yields.

Source	Sum of Squares	Degrees of freedom	Mean square	F-value	p-value
Model	5928.25	9	658.69	1820.99	< 0.0001
X_1	686.39	1	686.39	1897.55	< 0.0001
X_2	195.66	1	195.66	540.92	< 0.0001
X_3	242.07	1	242.07	669.20	< 0.0001
X_1X_2	77.48	1	77.48	214.21	< 0.0001
X_1X_3	47.66	1	47.66	131.75	< 0.0001
X_2X_3	94.70	1	94.70	261.81	< 0.0001
X_1^2	123.43	1	123.43	341.24	< 0.0001
X_2^2	4241.52	1	4241.52	11725.87	< 0.0001
X_3^2	26.14	1	26.14	72.27	0.0004
Residual	1.81	5	0.3617		
Lack of Fit	1.70	3	0.5670	10.53	0.0880
Pure Error	0.1077	2	0.0538		
Cor Total	5930.06	14			

Figure 6. Correlation graph of predicted values and actual yields of total flavonoid compounds

(a)

(b)

(c)

Figure 7. 3D response surface graphs of (a) the flavonoid yield versus extraction time (x1) and water content (x2); (b) flavonoid yield versus extraction time (x1) and solid-to-liquid ratio (x3); (c) flavonoid yield versus water content (x2) and solid-to-liquid ratio (x3).

4.3. Comparison with a conventional extraction solvent

To investigate the efficiency of DES as an alternative solvent for the extraction of flavonoids from the *I. javanica* flower, the extract obtained under optimum conditions was compared with the ethanolic extract. Under the same conditions, ethanol, as a conventional organic solvent, provided significantly lower total flavonoid compound quantities than DES (shown in Figure 8). Thus, it can be concluded that DES has better extraction efficiency than ethanol.

Figure 8. Comparison of the total flavonoid yields from DES and ethanol

4.4. SEM analysis

The outer surface of the *I. javanica* flower after the extraction process was captured and compared with the outer surface before extraction (shown in Figure 9-10). Figure 9a and 10a show that the surface of the *I. javanica* flower particle was compact, undamaged, and had no porosity. This structure changed after extraction using the UAE method. Many studies reported that ultrasound cavitation impacts the surface of solid plant materials, which subsequently becomes the mechanism leading to extraction enhancements in UAE, such as fragmentation, erosion, the sonocapillary effect, detexturation, local shear stress, and sonoporation [52-53].

Figures 9c, 9e, 10c, and 10e show that the surface of the *I. javanica* flower material after extraction using DES became more porous compared to that using ethanol (Figure 9b and 10b). Increasing the porosity increases the diffusion process and mass transfer.

As mentioned previously, because of the higher choline chloride content in DES, flavonoid yields were decreased due to the higher viscosity. In a high viscosity environment, the microbubble formation in the cavitation phenomenon will be disrupted, thereby decreasing the extraction efficiency [54,55]. Figure 9d and 10d show that there were only slight changes in porosity compared to the outer surface before the extraction process.

45 **Figure 9.** SEM image at a magnification of 500x of *I. javanica* flower powder (a) before the extraction process and after extraction
46 using (b) ethanol, (c) a DES combination of ChCl-Pg (1:1), (d) a DES combination of ChCl-Pg (2:1), and (e) a DES combination of
47 ChCl-Pg (1:2)

48
49
50
51
52
53
54
55
56
57
58
59
60

Figure 10. SEM image at a magnification of 5000x of *I. javanica* flower powder (a) before extraction process and after extraction using (b) ethanol, (c) a DES combination of ChCl-Pg (1:1), (d) a DES combination of ChCl-Pg (2:1), and (e) a DES combination of ChCl-Pg (1:2)

5. Conclusion

A combination of the UAE method  DES is a promising alternative environmentally friendly extraction method for obtaining high flavonoid yields from the *I. javanica* flower. The use of DES as an extraction solvent not only decreased the side effects of toxic organic solvent but also increased the extraction efficiency compared to a conventional solvent extraction. The DES used in this study was choline chloride–propylene glycol at a molar ratio of 1:1. Different molar ratio and viscosity values of DES, as well as water content, extraction time, and solid-to-liquid ratio values influenced the total flavonoid yields.

In the present study, the optimum ultrasound-assisted deep eutectic solvent extraction conditions suggested by the response surface analysis were an extraction time of 40 min, 25% water content in DES, and a solid-to-liquid ratio of 1:25 g/mL. The SEM image shows that the efficiency of extraction correlates to the porosity formation of the outer surface of the *I. javanica* flower.

Data accessibility

The data supporting the results in this article can be accessed at the Dryad Digital Repository: <https://doi.org/10.5061/dryad.fbg79cns9>

Funding

The authors appreciate the financial support provided by The Ministry of Research, Technology and Higher Education of the Republic of Indonesia for PDUPT 2020 Grant Ref number NKB-86/UN2.RST/HKP.05.00/2020. Nina Dewi Oktaviyanti was also supported by the Universitas Surabaya, Indonesia, with a PhD program scholarship.

Competing Interests

The authors declare no competing interest

References

- Marchand LL. 2002 Cancer preventive effects of flavonoids - a review. *Biomed Pharmacother.* **56**, 296–301. (doi:10.1016/s0753-3322(02)00186-5)
- Routray W, Orsat V. 2012 Microwave-Assisted Extraction of Flavonoids: A Review. *Food Bioprocess Tech.* **5** (2), 409–424. (doi:10.1007/s11947-011-0573-z)
- Jing C, Dong X, Tong J. 2015 Optimization of Ultrasonic-Assisted Extraction of Flavonoid Compounds and Antioxidants from Alfalfa Using Response Surface Method. *Molecules.* **29**, 15550–15571. (doi:10.3390/molecules20091550)
- Huang R, Wu W, Shen S, Fan J, Chang Y, Chen S, Ye X. 2018 Evaluation of colorimetric methods for quantification of citrus flavonoids to avoid misuse. *Anal. Methods.* **3**, 2575–2587. (doi:10.1039/c8ay00661j)
- Jalili-baleh L, Babaei E, Abdpour S, Nasir S, Bukhari A, Foroumadi A, Ramazani A, Sharifzadeh M, Abdollahi M, Khoobi M. 2018 A review on flavonoid-based scaffolds as multi-target-directed ligands (MTDLs) for Alzheimer's disease. *Eur. J. Med. Chem.* **152**, 570–589. (doi:10.1016/j.ejmech.2018.05.04)
- Sun Y, Wang W. 2008 Ultrasonic extraction of ferulic acid from *Ligusticum chuanxiong*. *J. Chin. Inst. Chem. Eng.* **39**, 653–656. (doi:10.1016/j.jcice.2008.05.012)
- Tiwari, B.K. 2015 Trends in Analytical Chemistry Ultrasound: A clean, green extraction technology. *Trends Anal. Chem.* **71**, 100–109. (doi:10.1016/j.trac.2015.04.013)
- Vinatoru M, Mason, TJ, Calinescu I. 2017 Ultrasonically assisted extraction (UAE) and microwave assisted extraction (MAE) of functional compounds from plant materials. *Trends Anal. Chem.* **97**, 159–178. (doi:10.1016/j.trac.2017.09.002)
- Ravanfar R, Moein M, Niakousari M, Tamaddon A. 2018 Extraction and fractionation of anthocyanins from red cabbage: ultrasonic-assisted extraction and conventional percolation method. *J. Food Meas. Charact.* **0** (0), 0. (doi:10.1007/s11694-018-9844-y)
- Jeong KM, Zhao J, Jin Y, Heo SR, Han SY, Yoo DE, Lee J. 2015 Highly efficient extraction of anthocyanins from grape skin using deep eutectic solvents as green and tunable media. *Arch. of Pharm. Res.* **38**, 2143–2152. (doi:10.1007/s12272-015-0678-4)
- Wei Z, Wang X, Peng X, Wang W, Zhao C. 2015 Fast and green extraction and separation of main bioactive flavonoids from *Radix Scutellariae*. *Ind. Crops Prod.* **63**, 175–181. (doi:10.1016/j.indcrop.2014.10.013)
- Li L, Liu JZ, Luo M, Wang W, Huang YY, Efferth T, Wang M, Fu YJ. 2016 Efficient extraction and preparative separation of four main isoflavonoids from *Dalbergia odorifera* T. Chen leaves by deep eutectic solvents-based negative pressure cavitation extraction followed by macroporous resin column chromatography. *J. Chromatogr. B.* **1033**, 40–48. (doi:10.1016/j.jchromb.2016.08.005)
- Bajkacz S, Adamek J. 2017 Evaluation of new natural deep eutectic solvents for the extraction of iso flavones from soy products. *Talanta.* **168** (February), 329–335. (doi:10.1016/j.talanta.2017.02.065)
- Meng Z, Jing Z, Hongxia D, Yuanyuan G, Longshan Z. 2018 Green and efficient extraction of four bioactive flavonoids from *Pollen Typhae* by ultrasound-assisted deep eutectic solvents extraction. *J. Pharm. Biomed.*

- 161, 246–253. (doi:10.1016/j.jpba.2018.08.048)
15. García G, Aparicio S, Ullah R, Atilhan M. 2015 Deep Eutectic Solvents: Physicochemical Properties and Gas Separation Applications. *Energ. Fuel*, **29**(4), 2616–2644. (doi:10.1021/ef5028873)
16. Liu Y, Friesen JB, Mcalpine JB, Lankin DC, Chen S, Pauli GF. 2018 Natural Deep Eutectic Solvents: Properties, Applications, and Perspectives. *J. Nat. Prod.* **81**, 679–690. (doi:10.1021/acs.jnatprod.7b00945)
17. Cunha SC, Fernandes O. 2018 Extraction techniques with deep eutectic solvents. *Trends Anal. Chem.* **105**. (doi:10.1016/j.trac.2018.05.001)
18. Dontha S, Hemalatha K, Mantripragada BR. 2015 Phytochemical and Pharmacological Profile of *Ixora*: A Review. *Int. J. Pharm. Sci. Res.* **6**(2), 567–584. (doi:10.13040/IJPSR.0975-8232.6(2).567-84)
19. Vishwanadham Y, Sunitha D, Ramesh A. 2016 Phytochemical Evaluation of Anti-Inflammatory Activity of Different Solvents Extracts of *Ixora javanica* Flowers. *Nat. Prod. Chem. Res.* **04**(03), 3–5. (doi:10.4172/2329-6836.1000219)
20. Nair SC, Panikkar B, Akamanchi KB, Panikkar KR. 1991 Inhibitory effect of *Ixora javanica* extract on skin carcinogenesis in mice & its antitumour activity. *Cancer Lett.* **60**, 253–258. (doi:10.1016/0304-3835(91)90121-W)
21. Hemalatha K, Darsini KP, Sunitha D. 2012 Hepatoprotective Activity of *Ixora javanica* D. C. Flowers against CCl₄ - induced Liver Damage in Rats. *Res. J. Pharm. Technol.* **5**(11), 1438–1441.
22. Rohini S, Shalimi M, Narayanaswamy N, Balakrishnan KP. 2012 Application of natural products in cosmetics: a study of *Ixora coccinea* extracts for their antityrosinase and antioxidant activities. *Int. J. Cosmet. Sci.* **2**(1), 1–7.
23. Dontha S, Kamurthy H, Mantripragada BR. 2016 Phytochemical screening and evaluation of in-vitro antioxidant activity of extracts of *Ixora javanica* D.C flowers. *Am. Chem. Sci. J.* **10**(1), 1–9. (doi:10.13040/IJPSR.0975-8232.6(2).567-84)
24. Oktaviyanti ND, Kartini, Mun'im A. 2019 Application and optimization of ultrasound-assisted deep eutectic solvent for the extraction of new skin-lightening cosmetic materials from *Ixora javanica* flower. *Heliyon.* **5**(October), e02950. (doi:10.1016/j.heliyon.2019.e02950)
25. Ruesgas-Ramon M, Figueroa-espinoza MC, Durand E. 2017 Application of Deep Eutectic Solvents (DES) for Phenolic Compounds Extraction: Overview, Challenges, and Opportunities. *J. Agric. Food Chem.* **65**, 3591–3601. (doi:10.1021/acs.jafc.7b01054)
26. Cao J, Wang H, Zhang W, Cao F, Ma G, Su E. 2018 Tailor-Made Deep Eutectic Solvents for Simultaneous Extraction of Five Aromatic Acids from *Ginkgo biloba* Leaves. *Molecules.* (doi:10.3390/molecules23123214)
27. Makoš P, Słupek E, Gębicki J. 2020 Hydrophobic deep eutectic solvents in microextraction techniques – A review. *Microchem. J.* **152**(October 2019), 104384. (doi:10.1016/j.microc.2019.104384)
28. Skarpalezos D, Detsi A. 2019 Review: Deep Eutectic Solvents as Extraction Media for valuable Flavonoids from Natural Sources. *Appl. Sci.* **9**, 4169. (doi:10.3390/app9194169)
29. Mulia K, Muhammad F, Krisanti E. 2017 Extraction of vitexin from binahong (*Anredera cordifolia* (Ten.) Steenis) leaves using betaine-1,4 butanediol natural deep eutectic solvent (NADES). *AIP Conf. Proc.* **020018**. (doi:10.1063/1.4978091)
30. Dai Y, Row KH. 2019 Application of Natural Deep Eutectic Solvents in the Extraction of Quercetin from Vegetables. *Molecules* **24**, 2300. (doi:10.3390/molecules24122300)
31. Paiva A, Craveiro R, Aroso I, Martins M, Reis RL, Duarte ARC. 2014 Natural Deep Eutectic Solvents - Solvents for the 21st Century. *ACS Sustainable Chem. Eng.* **2**, 1063–1071. (doi:10.1002/chin.201424290)
32. Georgantzi C, Lioliou AE, Paterakis N, Makris D. 2017 Combination of Lactic Acid-Based Deep Eutectic Solvents (DES) with β -Cyclodextrin: Performance Screening Using Ultrasound-Assisted Extraction of Polyphenols from Selected Native Greek Medicinal Plants. *Agronomy*, **7**(3), 54. (doi:10.3390/agronomy7030054)
33. Zainal-abidin MH, Hayyan M, Hayyan A. 2017 New horizons in the extraction of bioactive compounds using deep eutectic solvents: A review. *Anal. Chim. Acta.* **979**, 1–23. (doi:10.1016/j.aca.2017.05.012)
34. Craveiro R, Aroso I, Flammia V, Carvalho T, Viciosa MT, Dionísio M, Barreiros S, Paiva A. 2016 Properties and thermal behavior of natural deep eutectic solvents. *J. Mol. Liq.* **215**, 534–540. (doi:10.1016/j.molliq.2016.01.038)
35. Radošević K, Čurko N, Gaurina Srček V, Cvjetko Bubalo M, Tomašević M, Kovačević Ganić K, Radojčić Redovniković I. 2016 Natural deep eutectic solvents as beneficial extractants for enhancement of plant extracts bioactivity. *LWT - Food Sci. Technol.* **73**, 45–51. (doi:10.1016/j.lwt.2016.05.037)
36. Dai Y, Rozema E, Verpoorte R, Choi YH. 2016 Application of natural deep eutectic solvents to the extraction of anthocyanins from *Catharanthus roseus* with high extractability and stability replacing conventional organic solvents. *J. Chromatogr. A.* **1434**, 50–56. (doi:10.1016/j.chroma.2016.01.037)
37. Li Z, Lee PI. 2016 Investigation on drug solubility enhancement using deep eutectic solvents and their derivatives. *Int. J. Pharm.* **505**, 283–288. (doi:10.1016/j.ijpharm.2016.04.018)
38. Bubalo MC, Curko N, Tomašević M, Ganić KK, Redovniković IR. 2016 Green extraction of grape skin phenolics by using deep eutectic solvents. *Food chem.* **200**, 159–166. (doi:10.1016/j.foodchem.2016.01.040)
39. Bosiljkov T, Dujmić F, Cvjetko Bubalo M, Hribar J, Vidrih R, Brnčić M, Zlatic E, Radojčić Redovniković I, Jokić S. 2017 Natural deep eutectic solvents and ultrasound-assisted extraction: Green approaches for extraction of wine lees anthocyanins. *Food Bioprod. Process.* **102**, 195–203. (doi:10.1016/j.fbp.2016.12.005)
40. Li G, Zhu T, Ho K. 2017 Isolation of Ferulic Acid from Wheat Bran with a Deep Eutectic Solvent and Modified Silica Gel. *Anal. Lett.* **50**, 12, 1926–1938. (doi:10.1080/00032719.2016.1261879)

41. Hammond OS, Bowron DT, Edler KJ. 2017 Deep Eutectic Solvents Very Important Paper the Effect of Water upon Deep Eutectic Solvent Nanostructure: An Unusual Transition from Ionic Mixture to Aqueous Solution. *Angew. Chem. Int. Ed.* **56**, 9782–9785. (doi:10.1002/anie.201702486)
42. Ozturk B, Parkinson C, Gonzalez-miquel M. 2018 Extraction of polyphenolic antioxidants from orange peel waste using deep eutectic solvents. *Sep. Purif. Technol.* **206**(May), 1–13. (doi:10.1016/j.seppur.2018.05.052)
43. Sang J, Li B, Huang Y, Ma Q, Liu K, Li C. 2018 Deep eutectic solvent-based extraction coupled with green two-dimensional HPLC-DAD-ESI-MS/MS for the determination of anthocyanins from *Lycium ruthenicum* Murr fruit. *Anal. Methods.* **10**, 1247–1257. (doi:10.1039/C8AY00101D)
44. Chong FC, Gwee XF. 2015 Ultrasonic extraction of anthocyanin from *Clitoria ternatea* flowers using response surface methodology. *Nat. Prod. Res.* **29**(15), 1485–1487. (doi:10.1080/14786419.2015.1027892)
45. Khezeli T, Daneshfar A, Sahraei R. 2016 A green ultrasonic-assisted liquid-liquid microextraction based on deep eutectic solvent for the HPLC-UV determination of ferulic, caffeic and cinnamic acid from olive, almond, sesame and cinnamon oil. *Talanta*, **150**, 577–585. (doi:10.1016/j.talanta.2015.12.077)
46. Syakfanaya AM, Saputri FC, Mun'im, A. 2019 Simultaneously Extraction of Caffeine and Chlorogenic Acid from *Coffea canephora* Bean using Natural Deep Eutectic Solvent-Based Ultrasonic Assisted Extraction. *Phcog. J.* **11**(2), 267–271. (doi:10.5530/pj.2019.11.41)
47. Suhaimi SH, Hasham R, Khairul M, Idris H, Ismail H.F, Ariffin NHM, Majid FAA. 2019 Optimization of Ultrasonic-Assisted Extraction Condition Followed by Solid Phase Extraction Fractionation from *Orthosiphon stamineus* Benth (Lamiace) Leaves for Antiproliferative Effect on Prostate Cancer Cells. *Molecules*, **24**, 4183. (doi:10.3390/molecules24224183)
48. Xie P, Huang L, Zhang C, You F. 2015 Reduced pressure extraction of oleuropein from olive leaves (*Olea europaea* L.) with ultrasound assistance. *Food Bioprod. Process.* **93**(November), 29–38. (doi:10.1016/j.fbp.2013.10.004)
49. Agcam E, Akyildiz A, Balasubramaniam VM. 2017 Optimization of anthocyanins extraction from black carrot pomace with thermosonication. *Food Chem.* **237**, 461–470. (doi:10.1016/j.foodchem.2017.05.098)
50. Xu D, Li Y, Meng X, Zhou T, Zhou Y, Zheng J, Zhang J, Li H. 2017 Natural Antioxidants in Foods and Medicinal Plants: Extraction, Assessment and Resources. *Int. J. Mol. Sci.* **18**, 20–31. (doi:10.3390/ijms18010096)
51. Casazza AA, Aliakbarian B, Mantegna S, Cravotto G, Perego P. 2010 Extraction of phenolics from *Vitis vinifera* wastes using non-conventional techniques. *J. Food Eng.* **100**(1), 50–55. (doi:10.1016/j.jfoodeng.2010.03.026)
52. Petigny L, Périno-issartier S, Wajsman J, Chemat F. 2013 Batch and Continuous Ultrasound Assisted Extraction of Boldo Leaves (*Peumus boldus* Mol.). *Int. J. Mol. Sci.* **14**, 5750–5764. (doi:10.3390/ijms14035750)
53. Chemat F, Rombaut N, Sicaire A, Meullemiestre A, Abert-vian M. 2017 Ultrasound assisted extraction of food and natural products. Mechanisms, techniques, combinations, protocols and applications. *Ultrason. Sonochem.* **34**(June 2016), 540–560. (doi:10.1016/j.ultsonch.2016.06.035)
54. Fang X, Gu S, Jin Z, Hao M, Yin Z, Wang J. 2018 Optimization of Ultrasonic-Assisted Simultaneous Extraction of Three Active Compounds from the Fruits of *Forsythia suspensa* and Comparison with Conventional Extraction Methods. *Molecules*. **23**, 2115. (doi:10.3390/molecules23092115)
55. Huang H, Xu Q, Belwal T, Li L, Aalim H, Wu Q, Duan Z, Zhang X, Luo Z. 2019 Ultrasonic impact on viscosity and extraction efficiency of polyethylene glycol: A greener approach for anthocyanins recovery from purple sweet potato. *Food chem.* **283**(January). (doi:10.1016/j.foodchem.2019.01.017)

Figure 1. Calibration curve of the quercetin standard

254x190mm (96 x 96 DPI)

Figure 2. The effect of different water quantities in DES on flavonoid yields

366x275mm (96 x 96 DPI)

Figure 3. The effect of different molar ratios of DES (a combination of choline chloride and propylene glycol) and DES viscosities on flavonoid yields

254x190mm (96 x 96 DPI)

1
2
3
4
5
6
7
8
9
10
11
12
13
14
15
16
17
18
19
20
21
22
23
24
25
26
27
28
29
30
31
32
33
34
35
36
37
38
39
40
41
42
43
44
45
46
47
48
49
50
51
52
53
54
55
56
57
58
59
60

Figure 4. The effect of different extraction times on flavonoid yields
254x190mm (96 x 96 DPI)

Figure 5. The effect of different solid-to-liquid ratios on flavonoid yields

254x190mm (96 x 96 DPI)

1
2
3
4
5
6
7
8
9
10
11
12
13
14
15
16
17
18
19
20
21
22
23
24
25
26
27
28
29
30
31
32
33
34
35
36
37
38
39
40
41
42
43
44
45
46
47
48
49
50
51
52
53
54
55
56
57
58
59
60

Figure 6. Correlation graph of predicted values and actual yields of total flavonoid compounds

254x190mm (96 x 96 DPI)

Figure 7. 3D response surface graphs of (a) the flavonoid yield versus extraction time (x1) and water content (x2); (b) flavonoid yield versus extraction time (x1) and solid-to-liquid ratio (x3); (c) flavonoid yield versus water content (x2) and solid-to-liquid ratio (x3).

190x275mm (96 x 96 DPI)

1
2
3
4
5
6
7
8
9
10
11
12
13
14
15
16
17
18
19
20
21
22
23
24
25
26
27
28
29
30
31
32
33
34
35
36
37
38
39
40
41
42
43
44
45
46
47
48
49
50
51
52
53
54
55
56
57
58
59
60

Figure 8. Comparison of the total flavonoid yields from DES and ethanol

254x190mm (96 x 96 DPI)

Figure 9. SEM image at a magnification of 500x of *I. javanica* flower powder (a) before the extraction process and after extraction using (b) ethanol, (c) a DES combination of ChCl-Pg (1:1), (d) a DES combination of ChCl-Pg (2:1), and (e) a DES combination of ChCl-Pg (1:2)

190x227mm (96 x 96 DPI)

45 Figure 10. SEM image at a magnification of 5000x of *I. javanica* flower powder (a) before extraction process
46 and after extraction using (b) ethanol, (c) a DES combination of ChCl-Pg (1:1), (d) a DES combination of
47 ChCl-Pg (2:1), and (e) a DES combination of ChCl-Pg (1:2)

48 189x228mm (96 x 96 DPI)

Table 1. The code, range, and level of each variable selected for the experimental design.

Variables	Unit	Code	Range and level (xi)		
			-1	0	1
Time	min	X ₁	20	30	40
Water content in DES	%	X ₂	15	25	35
Solid-to-liquid ratio	g/mL	X ₃	1/25	1/26	1/27

Table 2. Experimental responses for different combination of variables.

Run	Independent variable			Response
	X ₁	X ₂	X ₃	Total flavonoid (mg QE/ g dried flowers)
1	-1	-1	0	35.292
2	1	-1	0	45.868
3	0	-1	-1	38.025
4	-1	1	0	16.389
5	-1	0	1	61.339
6	-1	0	-1	65.156
7	0	1	-1	38.075
8	1	1	0	44.57
9	0	0	0	63.527
10	0	0	0	63.908
11	0	-1	1	36.474
12	1	0	1	72.108
13	0	1	1	17.061
14	1	0	-1	89.732
15	0	0	0	63.488

Table 3. Analysis of variance for the prediction model of flavonoid yields.

Source	Sum of Squares	Degrees of freedom	Mean Square	F-value	p-value
Model	5928.25	9	658.69	1820.99	< 0.0001
X ₁	686.39	1	686.39	1897.55	< 0.0001
X ₂	195.66	1	195.66	540.92	< 0.0001
X ₃	242.07	1	242.07	669.20	< 0.0001
X ₁ X ₂	77.48	1	77.48	214.21	< 0.0001
X ₁ X ₃	47.66	1	47.66	131.75	< 0.0001
X ₂ X ₃	94.70	1	94.70	261.81	< 0.0001
X ₁ ²	123.43	1	123.43	341.24	< 0.0001
X ₂ ²	4241.52	1	4241.52	11725.87	< 0.0001
X ₃ ²	26.14	1	26.14	72.27	0.0004
Residual	1.81	5	0.3617		
Lack of Fit	1.70	3	0.5670	10.53	0.0880
Pure Error	0.1077	2	0.0538		
Cor Total	5930.06	14			

Appendix B

August 19, 2020

Dr. Laura Smith

Publishing Editor, Journals

Dear Editor,

Thank you for the opportunity to revise our manuscript entitled “A Green Extraction Design for Enhancing Flavonoid Compounds from the *Ixora javanica* flowers using a Deep Eutectic Solvent”. We greatly appreciate all the positive and constructive comment, review and suggestions given to us. We believe that the manuscript is substantially improved after making the suggestions. Following this letter, we also enclose our responses of all the comment from the reviewer in italics, including the modified texts and responses. Thank you for your consideration.

Sincerely

Nina Dewi Oktaviyanti

REVIEWER #1

1. Title: flowers instead of flower.

*Thank you for the suggestion. As suggested, we have made a change for the title become “A Green Extraction Design for Enhancing Flavonoid Compounds from the *Ixora javanica* flowers using a Deep Eutectic Solvent”*

2. Abstract: Can be formulated shorter without losing information. Add why flavonoids in *I. javanica* are so important.

*Thank you for the suggestion. We have tried hard to formulate the abstract to be shorter. The revised abstract contains all information that we think readers need to know. In addition, we also have added the reason why should be flavonoid in *I. javanica* in first sentence.*

3. Introduction: First sentence: biological activities in plants? This is followed by biological activities in humans.

Thank you, it has been revised as suggested.

4. Transfer information from the one but last paragraph to the Discussion and align with the text there. It contains too much details for an introduction.

Thank you, as suggested, we have deleted this part from introduction and we have moved and merged it into 4.1.1 section (first paragraph, sentence 9-13):

“Theoretically, the desirable extraction temperature using DES ranges from room temperature to about 60 °C [28]. The extractionprovided the maximum extraction recovery for flavonoid compounds”

5. Check legends of figures and tables. They are often confusing, lack relevant information.
Table 3: x-axis molar ratio of what?

Do you mean figure 3? x-axis of figure 3 (now figure 2) shows molar ratio of DES. We have made correction to all figures and tables.

6. I miss statistical evaluation of extraction time.

Thank you, in our revised manuscript, we have inserted statistical evaluation in fourth sentence (section 4.1.2).

7. 4.2: refer to Table 1 as well.

Section 4.2 should be rewritten and reorganized and better aligned with the Methods section. Explain the BB design to the reader.

Exactly, the extraction variables mentioned in section 4.2 refer to table 1. In our revised manuscript, we have described it in the third sentence of the first paragraph (section 4.2).

Thank you for the suggestion, we have made a lot of improvements in section 3.6 including explaining BB design.

8. Figure 6: add units.

Thank you for your suggestion. Figure 6 (now figure 5) is a graph of the observed (actual) response values versus the predicted response values. It helps to detect whether observations can be predicted well by the model. Indeed, this graph does not use units on the x or y axes.

9. A long extraction results loss of flavonoid content. What happens? You measure the total content calculated as quercetin. Add temperature measurements of the extraction mixture as a function of time.

We have revised our previous statement explained that flavonoid loss caused by increasing extraction time related temperature. We just realized that this was contrary with our extraction procedure which was maintained at room temperature.

10. In 4.4 the authors talk about 'porosity'. Please define this in the context of your work.

Thank you for the suggestion, we have tried to use different term to describe damage of the I. javanica surface. In addition, we have made a lot of improvements in section 4.4.

11. The authors present the extraction as 'green'. Of course, green is hot and sells, but work this out a bit better in the discussion. An important aspect, not mentioned by you, is how to get rid of the solvent. Ethanol easily evaporated, but the mixtures you used? How are the extracts applied in herbal medicine? This would place the work in a broader perspective. What do you want to achieve?

Recently, separating of DES as well as recyclability of DES is still a big challenge. From another point of view, we saw it as an opportunity. In this study, we did not do the DES separation on purpose. The evaporation process itself requires certain instruments and consumes energy. The reduction of this step can be used as a strategy in green extraction application.

“The use of DESs as a green solvent in the extraction process is currently very prospective because DESs are very simple, cheap, less toxic, have easy preparation, and can be adjusted according to the purpose of extraction [9-10]. Moreover, some HBA and HBD are also excipients in formulation of medicinal and cosmetic. Directly formulation of extract containing DES without solvent separation process can be used as a strategy to reduce one of energy-consuming process and excipient used.”

12. Figure 7 looks nice, but is hardly explained in the text. Please do so. Take the reader by the hand and lead him/her along your results. This can be done better!

Thank you for the suggestion. We have made a lot of improvements in section 4.2 including figure 7 (now figure 6) explanation (paragraph 3).

13. The conclusion is rather a repetition of the summary than a real conclusion. Make it shorter and more to the point.

As suggested, we have shorten the conclusion and made it different from abstract.

14. I recommend to reduce the number of references. A total of 55 is a lot for a study like this. Confine to the most relevant ones, it is not a review.

Thank you, we have tried to reduce the number of references from 55 to 47. We hope this is acceptable.

REVIEWER #2

1. Many formats about abbreviations and figures are not uniform in the manuscript. It should be carefully checked and revised. For instance, “*I. javanica*” (Page 2, lines 42 and 43) should be italic; “P” (Page 5, line 3) represents the abbreviation of propylene glycol, it look not proper; The ordinate scale in Figures 1, 4, 5 should be given as displayed in Figures 2 and 3.

Thank you, as suggested, we have checked and revised all the format and abbreviations in the manuscript.

We have decided to use full term of “propylene glycol” and “choline chloride” in the text, instead abbreviation. However, we still mention the combination of both compounds as abbreviation.

“The DES components used in this study were combination of choline chloride and propylene glycol (ChCl-Pg)”

2. The detailed reaction conditions except for corresponding variable in Figures 2, 3, 4, 5 should be given.

Thank you for the suggestion. Actually, we already described the extraction condition in section 3.4 extraction procedures.

“To determine the effect of water content in DES on the flavonoid yields, 0.5 g dried powder was extracted with 10 mL DES (ChCl-Pg at a molar ratio of 1:1), which contained different concentrations of water in DES (5%, 10%, 15%, 20%, 25%, 30%, and 35% w/w) for 5 min. Extraction time: 0.5 g dried powder was mixed with 10 mL DES (ChCl-Pg at a molar ratio of 1:1 and water content of 20% w/w), and the extraction was carried out under different extraction times (10, 15, 20, 25, 30, 35, and 40 min). The solid-to-liquid ratio: 0.5 g dried powder was mixed with different volumes of DES (ChCl-Pg at a molar ratio of 1:1 and water content of 20% w/w) until reaching a certain solid-to-liquid ratio (1:23, 1:24, 1:25, 1:26, 1:27, 1:28, and 1:29 g/ml) and was then sonicated for 5 min.”

3. In the section of 4.1 (Page 4, line 48), the authors discussed the variation trend of flavonoid yields with change of different extraction variables, but the detailed values of flavonoid yields are not mentioned.

Thank you, the values of flavonoid yields of each variables (section 4.1.1, 4.1.2, 4.1.3) have been added as suggested.

4. In order to make reader better understand the meaning of mathematical equation of the model (Page 7, line 10), its source should be given.

Thank you for the suggestion. Afore mentioned, we have made a lot of improvements in section 4.2. The mathematical equation is a second order polynomial model which expressed total flavonoid yields and it obtained from Design expert software. As suggested, we have stated this before mentioned the equation. In addition, this explanation also described in methods section (section 3.6).

5. This paper described the extraction efficiency of flavonoid from *Ixora javanica* flower using DES is superior to conventional extraction (ethanol), indicating DES is an effective method for flavonoid extraction. As we know, ethanol solvent is easy to recycle for next utilization, so the authors should consider the recyclability of DES in this manuscript.

Recently, separating of DES as well as recyclability of DES is still a big challenge. It is difficult to separate DES from the extracts. From another point of view, we saw it as an opportunity. In this study, we did not do the DES separation on purpose. The evaporation process itself requires certain instruments and consumes energy. The reduction of this step can be used as a strategy in green extraction application. We have described it in the second paragraph of introduction section:

“The use of DESs as a green solvent in the extraction process is currently very prospective because DESs are very simple, cheap, less toxic, have easy preparation, and can be adjusted according to the purpose of extraction [9-10]. Moreover, some HBA and HBD are also excipients in formulation of medicinal and cosmetic. Directly formulation of extract containing DES without solvent separation process can be used as a strategy to reduce one of energy-consuming process and excipient used.”